# Embryo polarity in moth flies and mosquitoes relies on distinct old genes with localized transcript isoforms

Yoseop Yoon[1], Jeff Klomp[1†], Ines Martin-Martin[2], Frank Criscione[2], Eric Calvo[2], Jose Ribeiro[2], Urs Schmidt-Ott[1*]

[1]Department of Organismal Biology and Anatomy, University of Chicago, Chicago, United States; [2]Laboratory of Malaria and Vector Research, National Institute of Allergy and Infectious Diseases, Rockville, United States

**Abstract** Unrelated genes establish head-to-tail polarity in embryos of different fly species, raising the question of how they evolve this function. We show that in moth flies (*Clogmia*, *Lutzomyia*), a maternal transcript isoform of *odd-paired (Zic)* is localized in the anterior egg and adopted the role of anterior determinant without essential protein change. Additionally, *Clogmia* lost maternal germ plasm, which contributes to embryo polarity in fruit flies (*Drosophila*). In culicine (*Culex*, Aedes) and anopheline mosquitoes (Anopheles), embryo polarity rests on a previously unnamed zinc finger gene (*cucoid*), or *pangolin* (*dTcf*), respectively. These genes also localize an alternative transcript isoform at the anterior egg pole. Basal-branching crane flies (*Nephrotoma*) also enrich maternal *pangolin* transcript at the anterior egg pole, suggesting that *pangolin* functioned as ancestral axis determinant in flies. In conclusion, flies evolved an unexpected diversity of anterior determinants, and alternative transcript isoforms with distinct expression can adopt fundamentally distinct developmental roles.

DOI: https://doi.org/10.7554/eLife.46711.001

**\*For correspondence:**
uschmid@uchicago.edu

**Present address:** [†]University of North Carolina, Lineberger Comprehensive Cancer Center, Chapel Hill, United States

**Competing interests:** The authors declare that no competing interests exist.

## Introduction

The specification of the primary axis (head-to-tail) in embryos of flies (Diptera) offers important advantages for studying how new essential gene functions evolve in early development. This process rests on lineage-specific maternal mRNAs that are localized at the anterior egg pole ('anterior determinants'), which, surprisingly, have changed during the evolution of flies. While the anterior determinants of most flies remain unknown, they can be identified by comparing the transcriptomes of anterior and posterior egg halves (*Klomp et al., 2015*). Furthermore, their function can be analyzed in the syncytial early embryos of a broad range of species via microinjection, considering timing and subcellular localization. It is therefore possible to conduct phylogenetic comparisons at the functional level. Finally, when the function of anterior determinants is suppressed, embryos develop into an unambiguous, predictable phenotype: these embryos lack all anterior structures and develop as two outward facing tail ends ('double abdomen').

Anterior determinants can be encoded by new genes with a dedicated function in establishing embryonic polarity. One example is *bicoid* in the fruit fly *Drosophila melanogaster*. Maternal mRNA of *bicoid* is localized in the anterior pole of the egg and Bicoid protein is expressed in a gradient in the early embryo (*Berleth et al., 1988*). Bicoid-deficient embryos fail to develop anterior structures and instead form a second tail end, or a symmetrical double abdomen when the maternal activity gradient of another gene, *hunchback,* is disrupted simultaneously (*Driever, 1993*). The *bicoid* gene originated in the lineage of cyclorrhaphan flies more than 140 million years ago by duplication of *zerknüllt* (*zen*; aka *Hox3*), which, in insects, plays an important role in extraembryonic tissue

**eLife digest** With very few exceptions, animals have 'head' and 'tail' ends that develop when they are an embryo. The genes involved in specifying these ends vary between species and even closely-related animals may use different genes for the same roles. For example, the products of two unrelated genes called *bicoid* in fruit flies and *panish* in common midges accumulate at one end of their respective eggs to distinguish head from tail ends. It remained unclear how other fly species, which have neither a *bicoid* nor a *panish* gene, distinguish the head from the tail end, or how genes can evolve the specific function of *bicoid* and *panish*.

Cells express genes by producing gene templates called messenger ribonucleic acids (or mRNAs for short). The central portions of mRNAs, known as protein-coding sequences, are then used to produce the protein. Proteins can play several distinct roles, which they acquire through evolution. This can happen in different ways, for example, genetic mutations in the part of a gene that codes for protein may alter the resulting protein, giving it a new activity. Alternatively, sequences at the beginning and the end of an mRNA molecule that do not code for protein, but regulate when and where proteins are made, can influence a protein's role by changing its environment. Many genes produce mRNAs with alternative sequences at the beginning or the end, a process known as alternative transcription.

Here, Yoon et al. identified three unrelated genes that perform similar roles to *bicoid* and *panish* in the embryos of several different moth flies and mosquitoes. These genes appear to have acquired their activity because one of their alternative transcripts accumulated at the future head end, rather than through mutations in the protein-coding sequences. Studying multiple species also made it clear that *panish* inherited its function from a localized alternative transcript of an old gene that duplicated and diverged.

These findings suggest that alternative transcription may provide opportunities for genes to evolve new roles in fundamental processes in flies. Most animal genes use alternative start and stop sites for transcription, but the reasons for this remain largely obscure. This is especially the case in the human brain. The findings of Yoon et al., therefore, raise the question of whether alternative transcription has played an important role in the evolution of the human brain.

DOI: https://doi.org/10.7554/eLife.46711.002

development (*Schmidt-Ott et al., 2010*). The expression and function of cyclorrhaphan *bicoid* orthologs are conserved but *bicoid* has not been found outside this group, and has been lost in some lineages within the Cyclorrhapha.

Another example is *panish*, which encodes the anterior determinant of a midge, *Chironomus riparius*. This gene evolved by gene duplication of the *Tcf* homolog *pangolin* (*pan*) and capture of the maternal promoter of a nucleoside kinase gene, and has been called *panish* (for *pan*"ish') (*Klomp et al., 2015*). Pangolin functions as the effector of ß-catenin-dependent Wnt signaling pathway ('canonical' Wnt signaling) but Panish lacks the ß-catenin domain of Pangolin, and sequence similarity between Pangolin and Panish is limited to the cysteine-clamp domain (30 amino acids). *panish* has not been found outside the family Chironomidae, suggesting that lower dipterans use different anterior determinants.

Here, we have used embryos of a wider range of dipteran species that lack *bicoid* and *panish* to address the question of how anterior determinants evolve. We started our analysis with moth flies (Psychodidae: *Clogmia albipunctata*, *Lutzomyia longipalpis*) and subsequently extended it to mosquitoes (Culicidae: *Culex quinquefasciatus*, *Aedes aegypti*, *Anopheles gambiae*, *Anopheles coluzzii*), and to crane flies (Tipulidae: *Nephrotoma suturalis*) (*Figure 1A*). Our results reveal three distinct old genes that evolved anterior determinants by localizing an alternative maternal transcript isoform at the anterior egg pole of the respective species. Therefore, alternative transcription might have played an important role in the evolution of this gene function and gene regulatory networks in fly embryos.

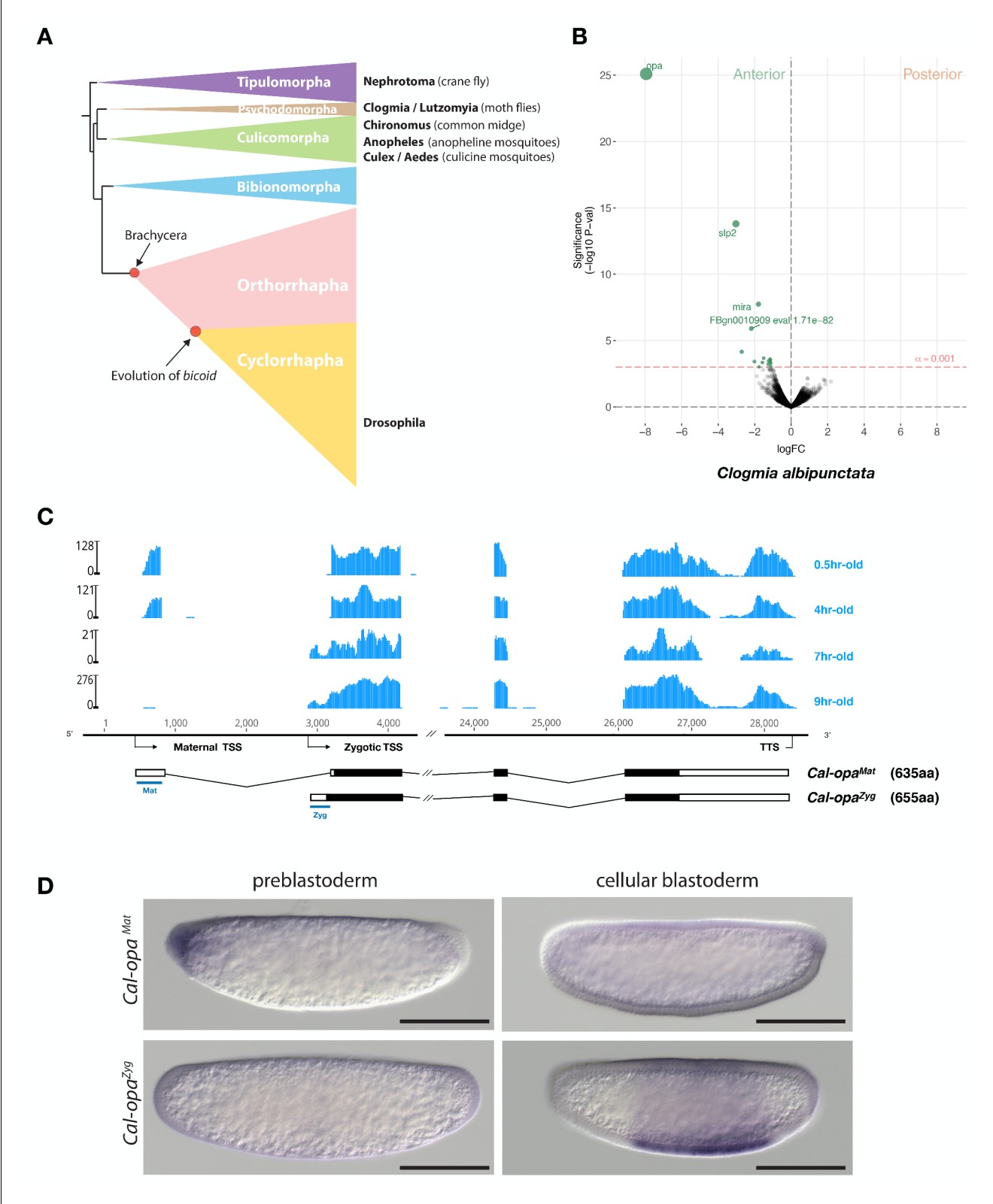

**Figure 1.** Expression of alternative *Cal-opa* transcripts in *Clogmia* embryos. (**A**) Phylogenetic relationship of fly species referred to in the text (***Wiegmann et al., 2011***). (**B**) Differential expression analysis of maternal transcripts between anterior and posterior halves of 1 hr-old *Clogmia* embryos. logFC: log fold-change. (**C**) Stage-specific RNA-seq read coverage of *Cal-opa* locus. Transcription start sites (TSS) and transcription termination sites (TTS) are indicated on the genomic scaffold (solid line with 1000 bp intervals marked) and were confirmed by RACE. Exon-intron sketches of *Cal-opa^Mat^*

*Figure 1 continued*

and *Cal-opa^{Zyg}* transcript variants are shown with the open reading frame in black and the position of in situ hybridization probes and dsRNA underlined in blue. (D) RNA in situ hybridization of *Cal-opa^{Mat}* and *Cal-opa^{Zyg}* transcripts in 1 hr-old preblastoderm and 7 hr-old cellular blastoderm embryos. Anterior is left and dorsal up. Scale bar: 100μm.

DOI: https://doi.org/10.7554/eLife.46711.003
The following figure supplement is available for figure 1:

**Figure supplement 1.** Protein alignment of predicted dipteran Odd-paired orthologs.
DOI: https://doi.org/10.7554/eLife.46711.004

## Results

### An alternative maternal transcript of the conserved segmentation gene *odd-paired* functions as anterior determinant in *Clogmia*

We annotated 5602 transcripts from the anterior and posterior transcriptomes of 1 hr-old bisected *Clogmia* embryos and ranked them according to the magnitude of their differential expression scores and P values (**Figure 1B**). In the anterior embryo, the most enriched transcript was homologous to *odd-paired*, the *Drosophila* homolog of mammalian *Zic* (*zinc finger of the cerebellum*) genes. ZIC proteins are known to function as transcription factors or co-factors (**Houtmeyers et al., 2013**). *odd-paired* was discovered in a screen for early *Drosophila* segmentation genes and subsequently classified as a 'pair-rule' gene, since *odd-paired* mutants fail to develop alternating segments (**Jürgens et al., 1984**). During the *Drosophila* segmentation process, *odd-paired* is expressed in a single broad domain and controls the 'frequency-doubling' of other pair-rule genes (**Clark and Akam, 2016**).

The *Clogmia* genome contains a single *odd-paired* locus (*Cal-opa*) (**Vicoso and Bachtrog, 2015**). Using RNA-seq data from preblastoderm and blastoderm embryos and Rapid Amplification of cDNA Ends (RACE), we identified maternal and zygotic *Cal-opa* transcripts with alternative first exons that we mapped onto a 54 kb genomic scaffold (**Figure 1C**). The maternal transcript (*Cal-opa^{Mat}*) was detected in preblastoderm embryos (0.5 hr-old) and syncytial blastoderm embryos (4 hr-old). The zygotic transcript (*Cal-opa^{Zyg}*) was found in cellularized blastoderm embryos (7 hr-old) and gastrulating embryos (9 hr-old). Protein alignments with homologs from other flies suggest that *Cal-opa^{Zyg}* encodes the full-length Cal-Opa protein (655 amino acids), while *Cal-opa^{Mat}* encodes a truncated protein variant (635 amino acids), lacking the N-terminal 20 amino acids of Cal-Opa^{Zyg} (**Figure 1—figure supplement 1**).

To confirm the alternative *Cal-opa^{Mat}* and *Cal-opa^{Zyg}* transcripts and their non-overlapping expression patterns, we performed whole mount RNA in situ hybridization experiments with transcript-specific probes. The *Cal-opa^{Mat}* transcript was anteriorly localized in preblastoderm embryos but absent at the cellular blastoderm stage. Conversely, the *Cal-opa^{Zyg}* transcript was absent in preblastoderm embryos but expressed broadly in the trunk region of 7 hr-old blastoderm embryos (**Figure 1D**), like *odd-paired* in *Drosophila*. These observations suggest that *Cal-opa* produces transcript isoforms with spatially and temporally distinct expression patterns.

To determine the function of *Cal-opa^{Mat}* and *Cal-opa^{Zyg}*, we established a protocol for microinjecting early *Clogmia* embryos and conducted transcript-specific RNA interference (RNAi) experiments. Injection of *Cal-opa^{Mat}* double-stranded RNA (dsRNA) led to mirror-image duplications of the tail end (double abdomen; **Figure 2A** and **Figure 2—figure supplement 1**). In contrast, injection of dsRNA targeting *Cal-opa^{Zyg}* resulted in half the number of segmental expression domains of *Cal-slp* (the ortholog of pair-rule gene *sloppy-paired*) and caused defects in segmentation, dorsal closure, and head development but did not alter embryo polarity (**Figure 2B**). Finally, injection of dsRNA targeting both *Cal-opa^{Mat}* and *Cal-opa^{Zyg}* resulted in double abdomens with missing segments (**Figure 2—figure supplement 2**). These observations indicate distinct roles of *Cal-opa^{Mat}* and *Cal-opa^{Zyg}* in specifying embryo polarity and in segmentation, respectively.

We noticed that maternal transcripts of *Cal-slp* and *Cal-mira*, a homolog of *miranda*, which encodes an adaptor protein for cell fate determinants in *Drosophila* (**Ikeshima-Kataoka et al., 1997; Adams et al., 2000**), were also slightly enriched in the anterior portion of embryo (**Figure 1B**). This observation was confirmed by RNA in situ hybridizations (**Figure 2—figure supplement 3**). Injection

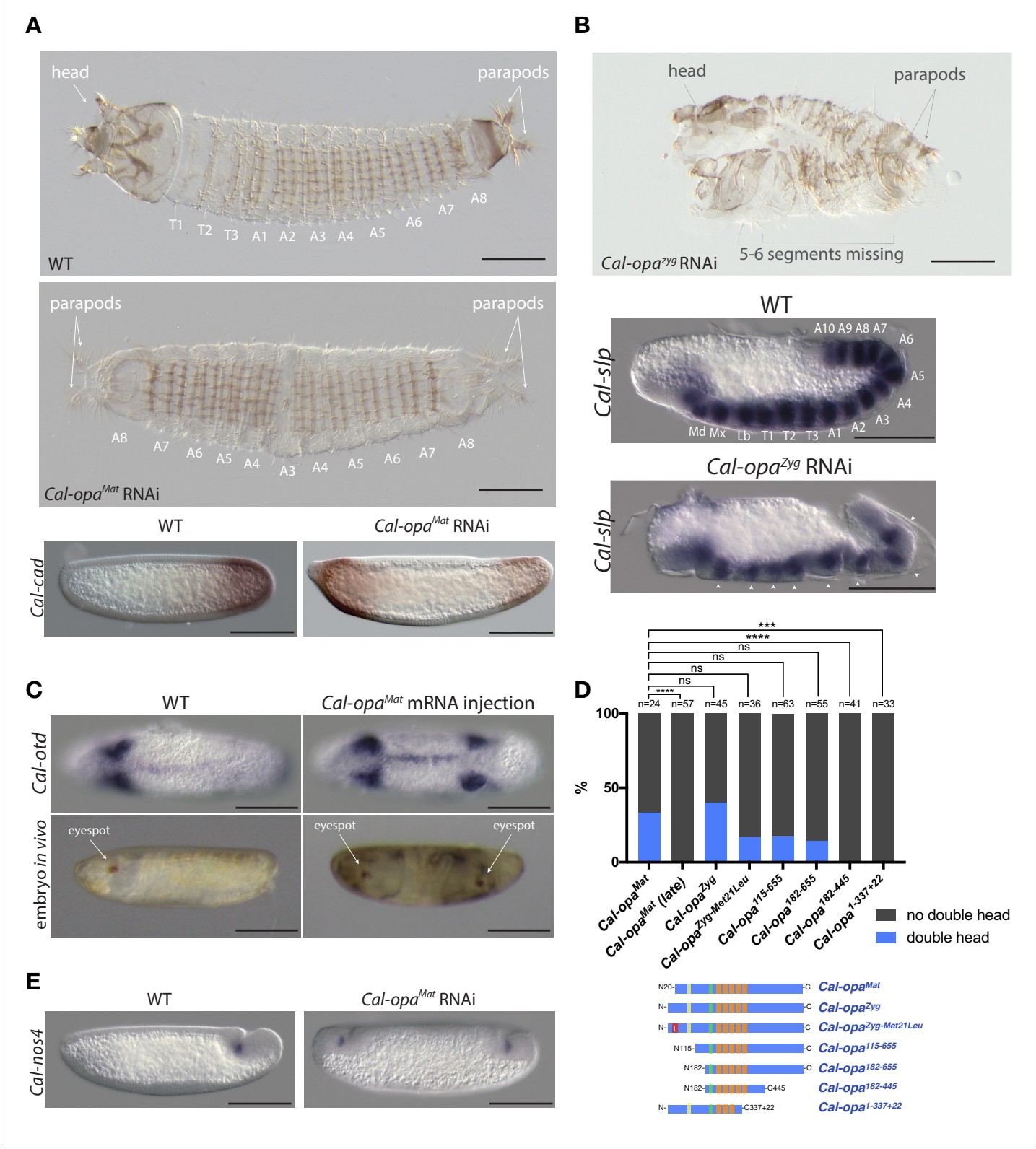

**Figure 2.** Function of alternative *Cal-opa* transcripts in early *Clogmia* embryo. (**A**) 1 st instar larval cuticle of wild type (top) and following *Cal-opa^Mat* RNAi (middle). RNA in situ hybridization of *Cal-cad* in a wild-type preblastoderm embryo (bottom left) and stage-matched *Cal-opa^Mat* RNAi embryos (bottom right). Anterior is left and dorsal up. T: thoracic segment; A: abdominal segment. Segment numbers in *Cal-opa^Mat* RNAi larval cuticle were assigned based on the assumption of polarity reversal. Scale bar: 100μm. (**B**) 1 st instar larval cuticle phenotype following *Cal-opa^Zyg* RNAi (top) and

*Figure 2 continued on next page*

*Figure 2 continued*

RNA in situ hybridization of *Cal-slp* in extending wild-type germband (middle) and stage-matched *Cal-opa^Zyg* RNAi embryo (bottom). Anterior is left and dorsal up. Md: mandibular segment; Mx: maxillary segment; Lb: labial segment; T: thoracic segment; A: abdominal segment. Scale bar: 100μm. (C) RNA in situ hybridizations of *Cal-otd* in wild-type gastrula (top left) and stage-matched embryo following posterior *Cal-opa^Mat* mRNA injection (top right) are shown in ventral view. A live wild-type embryo (bottom left) and a stage-matched embryo following posterior *Cal-opa^Mat* mRNA injection (bottom right) in lateral view. Anterior is left. Scale bar: 100μm. (D) Posterior injection of *Cal-opa* mRNA and mutated variants. Complete, symmetrical duplication of the bilateral *Cal-otd* expression domain in gastrulating embryos was counted as double head (blue, see *Figure 2C*). All other phenotypes, including incomplete duplications and wild type, were conservatively counted as 'no double head' (black). Sketches of predicted Cal-Opa proteins are shown with ZIC/Opa conserved motif (ZOC) in yellow, the ZIC family protein N-terminal conserved domain (ZFNC) in green, and zinc finger domains in orange. The Met21Leu mutation in Cal-Opa^Zyg-Met21Leu is marked in red. *Cal-opa^Mat* (late): *Cal-opa^Mat* was injected during the syncytial blastoderm stage (4 hr). ns: p>0.05; ***: p<0.001; ****: p<0.0001, Fisher's exact test. (E) RNA in situ hybridization of *Cal-nos4* in a gastrulating embryo (left) and stage-matched *Cal-opa^Mat* RNAi embryos (right).

DOI: https://doi.org/10.7554/eLife.46711.005

The following figure supplements are available for figure 2:

**Figure supplement 1.** Frequencies of strong *Cal-opa^Mat* and *Cal-opa^Zyg* RNAi phenotypes.

DOI: https://doi.org/10.7554/eLife.46711.006

**Figure supplement 2.** Cuticle phenotype of a 1 st instar larva following *Cal-opa^Mat* and *Cal-opa^Zyg* double RNAi.

DOI: https://doi.org/10.7554/eLife.46711.007

**Figure supplement 3.** Maternal transcript localization and RNAi phenotypes of *Cal-slp* and *Cal-mira* in *Clogmia*.

DOI: https://doi.org/10.7554/eLife.46711.008

**Figure supplement 4.** Cuticle phenotype of a 1 st instar *Clogmia* larva following *Cal-opa^Mat* mRNA injection.

DOI: https://doi.org/10.7554/eLife.46711.009

**Figure supplement 5.** *Clogmia* homologs of *nanos*, *vasa*, *tudor*, and *germ cell-less*.

DOI: https://doi.org/10.7554/eLife.46711.010

of *Cal-slp* dsRNA resulted in head and dorsal closure defects while *Cal-mira* dsRNA caused labrum and antennal defects, but in both cases embryo polarity was retained (*Figure 2—figure supplement 3*).

To test whether *Cal-opa^Mat* can induce head development ectopically, we injected *Cal-opa^Mat* mRNA into the posterior pole of 1 hr-old embryos. These embryos expressed a head marker, *Cal-otd* (ortholog of *ocelliless/orthodenticle*), on both ends of the embryo and developed a symmetrical double head, including some duplicated thoracic elements (*Figure 2C* and *Figure 2—figure supplement 4* and *Video 1*). These observations suggest that anterior enrichment of maternal transcripts other than *Cal-opa^Mat* mRNA is not essential for head development, and that *Cal-opa^Mat* localization is sufficient for establishing embryo polarity.

## The anterior determinant function of *Cal-opa* is sensitive to expression timing but insensitive to 5' truncation of the open reading frame

Next, we asked whether the early timing of *odd-paired* expression is critical for its function as anterior determinant in moth flies. To test this hypothesis, we conducted posterior injections of *Cal-opa^Mat* mRNA during the syncytial blastoderm stage (4 hr) and examined *Cal-otd* expression after gastrulation. These embryos developed with normal head-to-tail polarity (*Figure 2D*). This result places the requirement of *odd-paired* for axis specification prior to the syncytial blastoderm stage and suggests that early timing of *odd-paired* activity is essential for its function as anterior determinant.

*Cal-opa^Mat* and *Cal-opa^Zyg* mRNAs not only differ in the timing of expression, but also differ in their 5'UTRs and predicted N-terminal protein sequences, as mentioned above (*Figure 1C* and *Figure 1—figure supplement 1*). To test

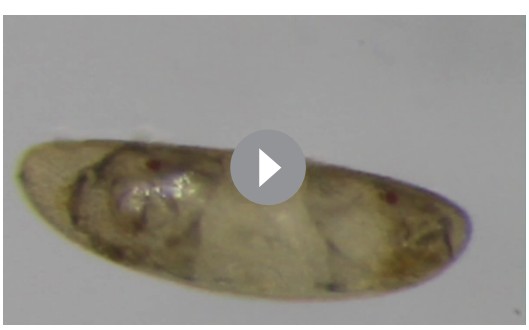

**Video 1.** Live double head embryo in ventrolateral view. The larval cuticle of this embryo is shown in *Figure 3—figure supplement 1*.

DOI: https://doi.org/10.7554/eLife.46711.011

whether the open reading frame difference is required for the anterior determinant function, we injected *Cal-opa*^Zyg^ mRNA at the posterior pole of preblastoderm embryos. Embryos from this experiment also developed as double heads (*Figure 2D*). Because the translation start site of Cal-Opa^Mat^ is located downstream of the Cal-Opa^Zyg^ translation start site, we also tested *Cal-opa*^Zyg^ mRNA in which the putative start codon for Cal-Opa^Mat^ was mutated to encode leucine (*Cal-opa*^Zyg-Met21Leu^). Posterior injection of *Cal-opa*^Zyg-Met21Leu^ mRNA also resulted in double heads (*Figure 2D*). These findings indicate that the protein difference between Cal-Opa^Mat^ and Cal-Opa^Zyg^ is not essential for the anterior determinant function of *odd-paired* in *Clogmia*.

To test whether only a small portion of the *Cal-opa* open reading frame is required for its function as anterior determinant, we examined the ability of various truncated variants of *Cal-Opa* mRNAs (*Figure 1—figure supplement 1*) to induce head development at the posterior egg pole (*Figure 2D*). mRNAs of protein variants with large N-terminal truncation (Cal-Opa^115-655^ and Cal-Opa^182-655^) retained the ability to induce double heads. However, mRNAs of protein variants with C-terminal truncation (Cal-Opa^182-445^ and Cal-Opa^1-337+22^, a hypothetical splice variant) failed to induce double heads. These results indicate that the ability of Cal-Opa to specify embryo polarity requires the C-terminal portion of the protein but is largely insensitive to N-terminal truncation, corroborating our above conclusion that N-terminal differences between Cal-Opa^Mat^ and Cal-Opa^Zyg^ were not essential for evolving the anterior determinant function of *Cal-opa*.

## *Cal-opa*^Mat^ suppresses zygotic germ cell specification at the anterior pole and *Clogmia* lacks maternal germ plasm

In *Drosophila* and other dipterans, maternal germ plasm in the posterior embryo not only specifies primordial germ cells but also contributes to and stabilizes embryo polarity via *nanos*, which suppresses the translation of anterior determinants in the posterior embryo (*Tautz, 1988*; *Gavis and Lehmann, 1992*; *Struhl et al., 1992*; *Lemke and Schmidt-Ott, 2009*). The activity of *nanos* in the posterior preblastoderm is dependent on *oskar* (*Lehmann, 2016*), which is conserved in many insects (*Ewen-Campen et al., 2010*). However, in *Clogmia*, expression profiling of anterior and posterior egg halves did not reveal any posteriorly localized maternal transcripts (*Figure 1B*, alpha = 0.001, unadj.), and no *oskar* homolog was found in our *Clogmia* transcriptomes or the *Clogmia* genome (*Vicoso and Bachtrog, 2015*). To test whether *Clogmia* lacks maternal germ plasm, we examined the expression of candidate germ cell markers, including *Clogmia* homologs of *nanos* (*Cal-nos1*, *Cal-nos2*, *Cal-nos3*, and *Cal-nos4*), *vasa* (*Cal-vas*), *tudor* (*Cal-tud*), and *germ cell-less* (*Cal-gcl*). *Cal-nos1*, *Cal-nos3*, and *Cal-nos4* were not localized in the posterior of preblastoderm embryos but were expressed in a small set of cells at the posterior pole of cellular blastoderm and gastrulating embryos along with *Cal-vas*, *Cal-tud*, and *Cal-gcl* that were expressed more broadly (*Figure 2—figure supplement 5*). These observations suggest that *Clogmia* lacks maternal germ plasm and that *Clogmia* may induce the germ cell fate zygotically. To test this hypothesis, we examined *Cal-nos* expression in *Cal-opa*^Mat^ RNAi embryos. *Cal-nos* positive cells were duplicated in double abdomens (*Figure 2E*), indicating that *Clogmia* uses an inductive mechanism for germ cell specification, which is repressed in the anterior embryo by *Cal-opa*^Mat^. Therefore, axis specification in the *Clogmia* embryo is independent from germ cell specification. To our knowledge, *Clogmia* is also the first example of inductive germ cell specification in flies.

## Evolution of the anterior determinant function of moth fly *odd-paired*

Maternal *odd-paired* transcript is absent in freshly deposited eggs of chironomids (*Klomp et al., 2015*) and mosquitoes (*Akbari et al., 2013*), both of which belong to the Culicomorpha lineage (*Figure 1A*). To test whether localized maternal *odd-paired* transcript is broadly conserved in the Psychodomorpha lineage, we examined maternal transcript localization in the eggs of the sand fly *Lutzomyia longipalpis*, a moth fly species of public health concern due to its role in the transmission of visceral leishmaniasis. Of 5392 annotated transcripts, the most enriched maternal transcript in the anterior half of 1–2 hr old embryos was homologous to *odd-paired* and was therefore named *Llo-opa*^Mat^ (*Figure 3A*). In the posterior *Lutzomyia* embryo, the most enriched transcript was homologous to *oskar*, indicating that *Lutzomyia* eggs contain maternal germ plasm at the posterior pole, unlike the *Clogmia* eggs. These findings suggest that a broad range of moth flies use *odd-paired* transcript as anterior determinant, and that maternal germ plasm was lost only in the *Clogmia*

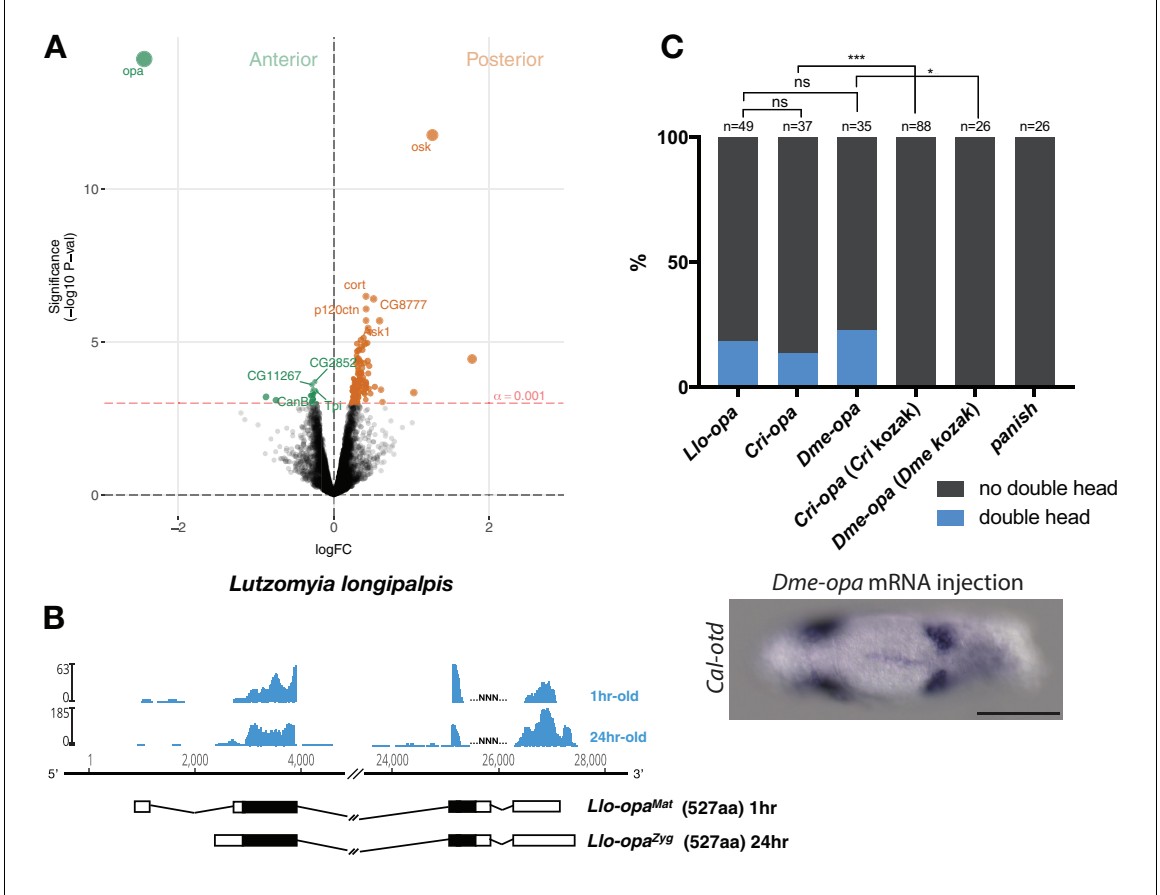

**Figure 3.** Alternative *odd-paired* transcript isoforms in *Lutzomyia* and ectopic head induction by mRNAs derived from *Cal-opa* orthologs in *Clogmia*. (A) Differential expression analysis of maternal transcripts between anterior and posterior halves of 1–2 hr-old *Lutzomyia* embryos. logFC: log fold-change. (B) Stage-specific RNA-seq read coverage of *Llo-opa* genomic locus and sketches of *Llo-opa*$^{Mat}$ and *Llo-opa*$^{Zyg}$ transcripts (see also legend to *Figure 1A*). (C) Posterior injection of *odd-paired* mRNAs from *Lutzomyia*, *Chironomus*, and *Drosophila* in *Clogmia* embryo. Phenotypes were counted as in (*Figure 2D*) and double head from *Dme-opa* mRNA injection is shown as example. *Cri-opa* and *Dme-opa* mRNAs include Kozak sequence of *Cal-opa*$^{Mat}$ (TAAG upstream of the predicted translation start site). *Cri*-kozak and *Dme*-kozak refer to *odd-paired* sequences with donor-specific Kozak-sequences from *Chironomus* (AAAA) and *Drosophila* (GACC), respectively. ns.: p>0.05; *: p<0.05. ***: p<0.001, Fisher's exact test.

DOI: https://doi.org/10.7554/eLife.46711.012

The following figure supplement is available for figure 3:

**Figure supplement 1.** Expression level of select *Lutzomyia* gap gene (red) and pair-rule gene (blue) homologs.

DOI: https://doi.org/10.7554/eLife.46711.013

lineage. Close examination of *Lutzomyia* transcriptomes from 1 hr-old and 24 hr-old embryos also revealed zygotic *odd-paired* transcript (*Llo-opa*$^{Zyg}$) (*Figure 3B*). *Llo-opa*$^{Mat}$ and *Llo-opa*$^{Zyg}$ share the same open reading frame but differ at their untranslated 5' and 3' ends. Since the N-terminal ends of Llo-Opa$^{Mat}$/Llo-Opa$^{Zyg}$ and Cal-Opa$^{Zyg}$ proteins are homologous (*Figure 1—figure supplement 1*), we infer that the N-terminal truncation of Cal-Opa$^{Mat}$ occurred after the transcript had evolved maternal expression and anterior localization. The detection of *Llo-opa*$^{Zyg}$ transcript in 24 hr-old embryos coincided with that of gap and pair-rule segmentation gene homologs (*Figure 3—figure supplement 1*), indicating that *Llo-opa*$^{Zyg}$ functions during segmentation.

The *odd-paired* gene of ancestral moth flies could have evolved the ability to establish the embryo polarity via specific amino acid substitutions. In this case, *odd-paired* homologs from species with a different anterior determinant, such as *Drosophila* or *Chironomus*, should not induce ectopic head development in *Clogmia* embryos. Alternatively, *odd-paired* could have evolved its role as axis determinant in moth flies independent of any amino acid substitution via co-option. In this case, *odd-paired* homologs from *Drosophila* or *Chironomus* could have the ability to induce head

development in *Clogmia* embryos when appropriately expressed. To test this possibility, we injected *odd-paired* mRNA from *Lutzomyia*, *Chironomus*, or *Drosophila* into the posterior pole of early *Clogima* embryos. All of these *odd-paired* homologs induced double heads in *Clogmia*, provided that the endogenous kozak sequence of *Cal-opa^Mat^* was used for optimal translation efficiency (*Figure 3C*). Since neither *Drosophila* nor *Chironomus* uses *odd-paired* for specifying embryo polarity, these results suggest that amino acid substitutions were not essential for the evolution of the anterior determinant function of *odd-paired* in moth flies. We therefore propose that this gene function evolved via co-option when alternative maternal transcript of moth fly *odd-paired* became enriched at the anterior egg pole.

## A previously uncharacterized C2H2 zinc finger gene, named *cucoid*, functions as anterior determinant in culicine mosquitoes

Given that freshly deposited mosquito eggs lack maternal *odd-paired* transcript orthologs (*Akbari et al., 2013*), we extended our search for anterior determinants to mosquitoes. Initially, we focused on the Southern House Mosquito *Culex quinquefasciatus*, a vector of West Nile virus (the leading cause of mosquito-borne disease in the continental United States) and of *Wuchereria bancrofti* (the major cause of lymphatic filariasis). This species was chosen because their eggs are large and have clearly distinguishable anterior and posterior egg poles. We annotated 8239 *Culex* transcripts from the pooled anterior and posterior transcriptomes of 1 hr-old preblastoderm embryos and ranked them according to the magnitude of their differential expression scores and P values (*Figure 4A*). In the posterior embryo, the most enriched transcript was related to *nanos*, consistent with the presence of maternal germ plasm in this species (*Figure 4D*) (*Juhn et al., 2008*). The most enriched transcript in the anterior embryo was closely related to an uncharacterized gene of *Drosophila* (*CG9215*). However, reciprocal BLAST searches suggest that *CG9215* belongs to a poorly defined larger gene family in *Drosophila* that might be represented by a single gene in mosquitoes. We named this gene *cucoid* to reflect its *bicoid*-like function in a culicine mosquito. *cucoid* encodes a protein with five C2H2 zinc finger domains (*Figure 4—figure supplement 1*). RACE experiments with cDNA from 0 to 7 hr-old embryos revealed three alternative *cucoid* transcripts with distinct 3' ends (*cucoid^A^, cucoid^B^,* and *cucoid^C^*) (*Figure 4B*), but only *cucoid^B^* and *cucoid^C^* were recovered from cDNA of 0–2 hr-old preblastoderm embryos, suggesting that one or both these transcripts might be maternally localized at the anterior pole. To test this hypothesis, we performed RNA in situ hybridizations with specific probes for *cucoid^A^* (probe A) or *cucoid^B^* (probe B) and, due to the very short sequence unique to *cucoid^C^* (121 nucleotides), a probe against all three isoforms (probe C). *cucoid^A^* and *cucoid^B^* expression was detected in the fore and hind gut of extended germbands but not in 1 hr-old preblastoderm embryos. In contrast, the probe against all three isoforms detected maternally localized *cucoid* transcript at the anterior pole in addition to the zygotic expression pattern (*Figure 4C*). Taken together, these results suggest that only the *cucoid^C^* isoform is maternally localized at the anterior pole and could function as anterior determinant. To test this hypothesis, we injected *cucoid* dsRNA from the shared 5' region and examined the expression of a posterior marker (*Cqu-cad*) in gastrulating embryos. Many of these embryos expressed *Cqu-cad* in the anterior and underwent ectopic gastrulation at the anterior pole, suggesting that normal head-to-tail polarity was lost (*Figure 4E*). Taken together, our results suggest that *cucoid* acquired the anterior determinant function via the localization of a maternal transcript isoform with an alternative 3' end.

We obtained similar results in another culicine mosquito, the Yellow Fever Mosquito *Aedes aegypti,* which transmits Dengue, Chikungunya, and Zika viruses. In this species, expression profiling of 5802 transcripts from the anterior and posterior transcriptomes of 1 hr-old preblastoderm embryos also identified *cucoid* (*Aae-cucoid*) as the gene with the most significantly enriched transcript in the anterior embryo (*Figure 5A*). RACE experiments with cDNA from 1 hr to 6 hr-old embryos revealed three similar *Aae-cucoid* transcripts with alternative 3' ends (*Aae-cucoid^A^, Aae-cucoid^B^,* and *Aae-cucoid^C^*) (*Figure 5B*), and RNA in situ hybridization experiments with a probe against all three isoforms confirmed the anterior localization of *Aae-cucoid* transcript in preblastoderm embryos (*Figure 5C*). *Aae-cucoid* expression in Aedes germbands could not be examined for technical reasons.

In the posterior embryo, no highly enriched transcripts were observed. This was unexpected given that whole mount in situ hybridizations revealed posterior localized transcript of Aedes *nanos* in 1 hr-old embryos (*Figure 5C*) and that maternal transcript of Aedes *oskar* is also localized at the

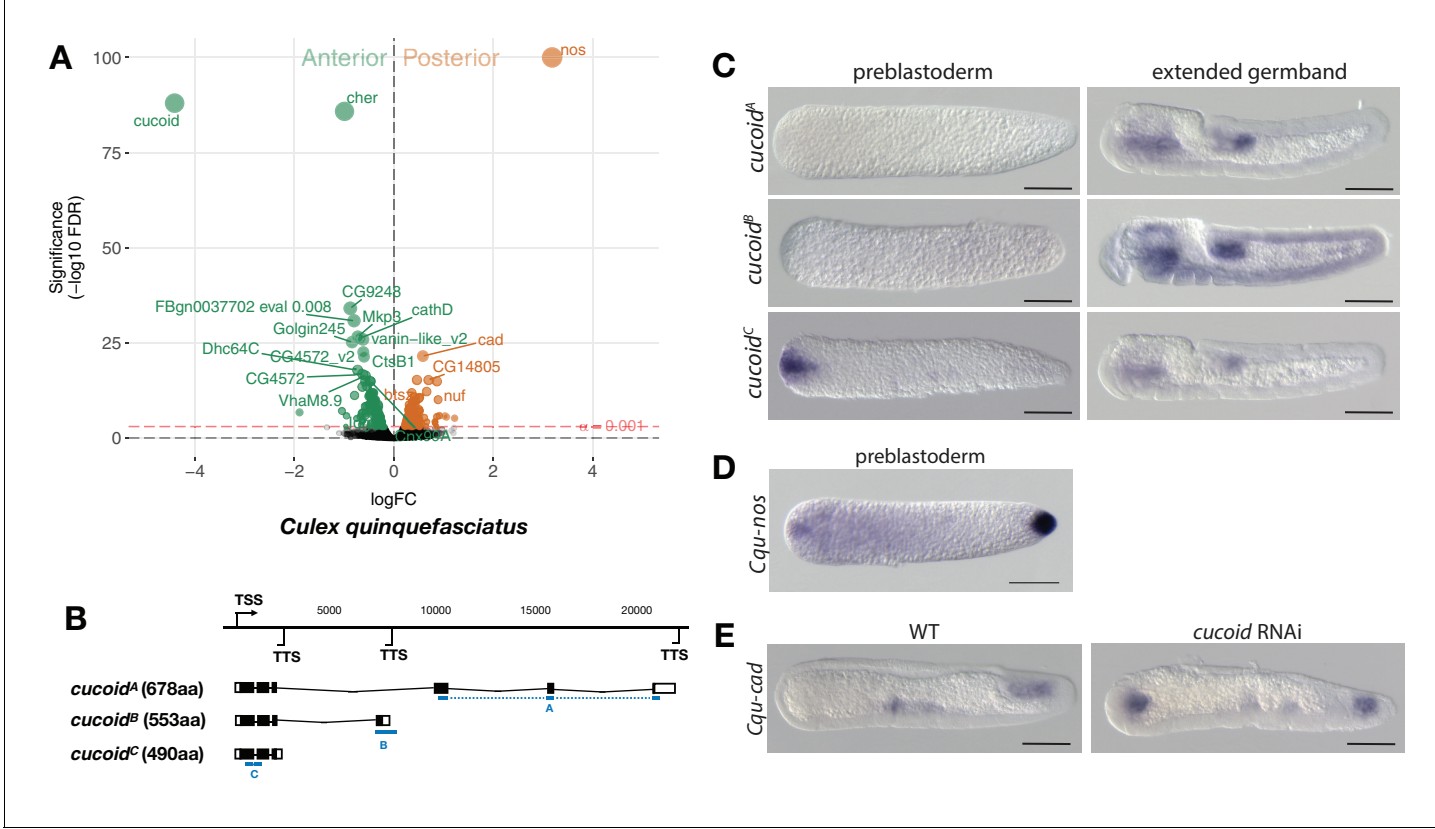

**Figure 4.** Expression and function of alternative *cucoid* isoforms in *Culex*. (A) Differential expression analysis of maternal transcripts between anterior and posterior halves of 1 hr-old *Culex* embryos. logFC: log fold-change. (B) Sketches of *cucoid* transcript isoforms based on RNA-seq and RACE experiments (see also legend to *Figure 1A*). Blue lines the position of in situ hybridization probes. (C) RNA in situ hybridization of *cucoid*[A], *cucoid*[B], and *cucoid*[C] transcripts in 1 hr-old preblastoderm and germband extending embryos. Anterior is left and dorsal up. Scale bar: 100μm. (D) RNA in situ hybridization of *Cqu-nos* in 1 hr-old preblastoderm embryo. Anterior is left. Scale bar: 100μm (E) RNA in situ hybridization of *Cqu-cad* in a wild-type gastrulating embryo (left) and in a stage-matched *cucoid* RNAi embryos (right; 16/48 with *cucoid* dsRNA versus 0/25 with lacZ control dsRNA; p<0.0005). Anterior is left and dorsal up. Scale bar: 100μm.

DOI: https://doi.org/10.7554/eLife.46711.014

The following figure supplement is available for figure 4:

**Figure supplement 1.** Protein alignment of dipteran *cucoid* orthologs and a *Drosophila* paralog CG4424.
DOI: https://doi.org/10.7554/eLife.46711.015

posterior pole (*Juhn and James, 2006*). Low statistical power of our differential expression analysis in Aedes might explain this discrepancy, since we could have confounded the anterior and posterior pole in some of the bisected Aedes eggs (see Materials and methods). Alternatively, only a small portion of these transcripts might be localized at the posterior pole. Injection of *Aae-cucoid* dsRNA against the shared region of all transcripts resulted in double abdomens (*Figure 5D*), suggesting that *cucoid* evolved its function as anterior determinant prior to the divergence of the *Culex* and *Aedes* lineages.

## *pangolin/Tcf* functions as anterior determinant in anopheline mosquitoes

The *Anopheles gambiae* species complex constitutes an outgroup to the *Culex-Aedes* clade. It includes eight or more sub-Saharan species that are difficult to distinguish due to widespread genealogical heterogeneity across the genome, incomplete lineage sorting and introgression (*Thawornwattana et al., 2018*). We interchangeably used *A. gambiae* and *A. coluzzii*, two sibling species within this species complex that are responsible for the majority of malaria transmission in Africa, to identify the anterior determinant of this mosquito lineage. Whole mount RNA in situ

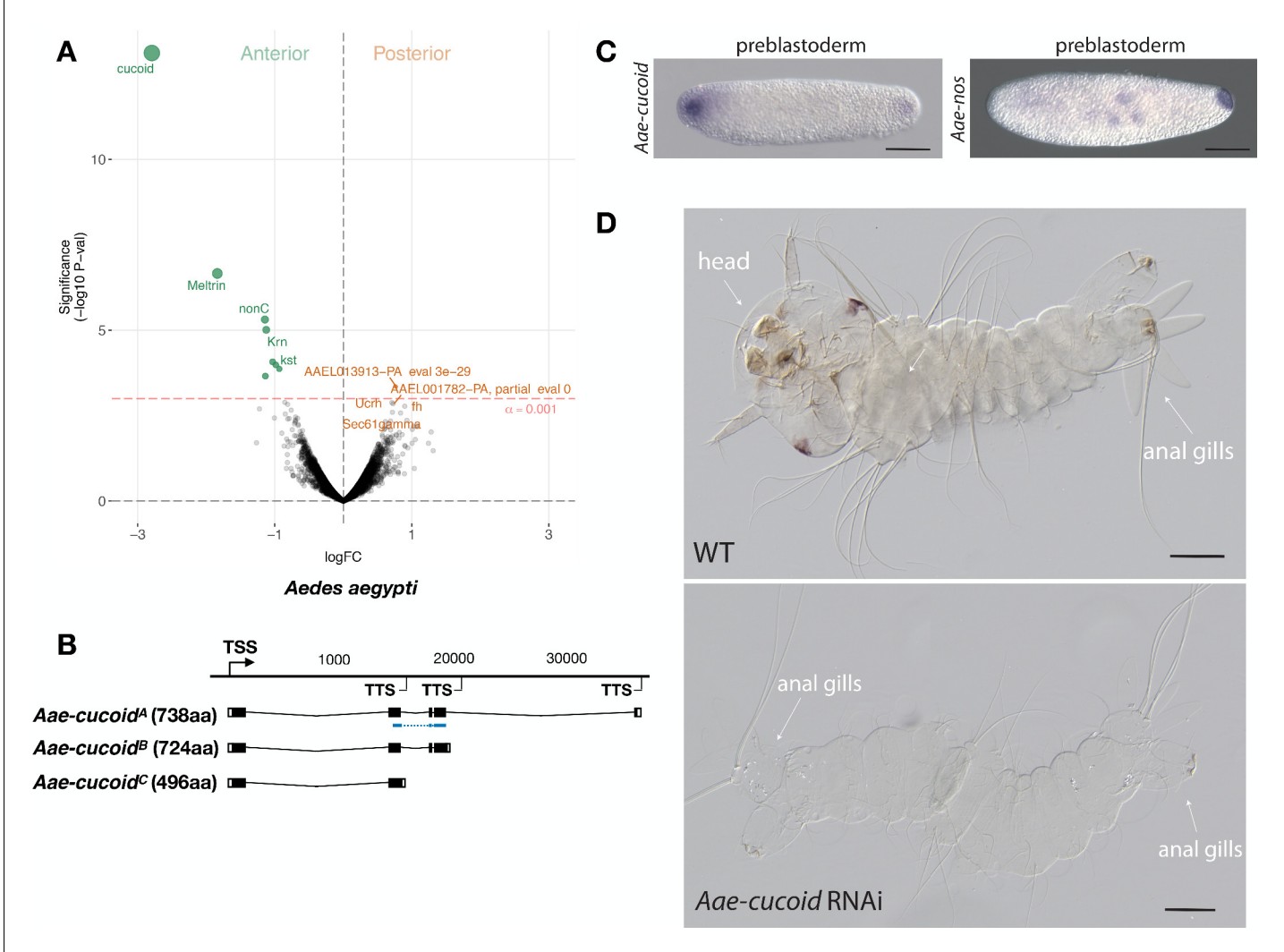

**Figure 5.** Expression and function *cucoid* in Aedes. (**A**) Differential expression analysis of maternal transcripts between anterior and posterior halves of 1 hr-old Aedes embryos. logFC: log fold-change. (**B**) Sketches of *Aae-cucoid* transcript isoforms based on RNA-seq and RACE experiments with the position of the in situ hybridization probe and dsRNA underlined in blue (see also legend to *Figure 1A*). (**C**) RNA in situ hybridization *Aae-cucoid* (left) and *Aae-nos* (right) transcript in 1 hr-old preblastoderm embryo. Anterior is left and dorsal up. Scale bar: 100μm. (**D**) 1 st instar Aedes larval cuticle of wild type (top) and following *Aae-cucoid* RNAi (bottom; 9/26 versus 0/22 with control dsRNA; p<0.005). Scale bar: 100μm.
DOI: https://doi.org/10.7554/eLife.46711.016

hybridizations with a probe against the *Anopheles gambiae* ortholog of *cucoid* did not detect any anterior localized transcript in 1 hr-old embryos, suggesting that *Anopheles* uses a different anterior determinant than *Culex* and *Aedes*.

To further test this possibility, we sequenced the anterior and posterior transcriptomes 1 hr-old preblastoderm embryos of *A. gambiae* and ranked 9353 transcripts according to the magnitude of their differential expression scores and P values. In the posterior embryo, the most enriched transcript was homologous to *nanos*. In the anterior embryo, the most enriched transcript was homologous to *pangolin* (also known as *Tcf*) (*Figure 6A*). To test for potential alternative maternal and zygotic isoforms of *pangolin* in *Anopheles*, we mapped the assembled transcripts and 5' and 3' RACE products from 1 to 6 hr-old embryos onto an available *A. gambiae* genome assembly (AgamP4). We identified two alternative transcript variants with non-overlapping 3'UTRs but nearly identical open reading frames that we named *Aga-pan^{Mat}* and *Aga-pan^{Zyg}*, respectively (*Figure 6B* and *Figure 6—figure supplement 1*). *Aga-pan^{Mat}* was tightly localized at the anterior pole of 1–2

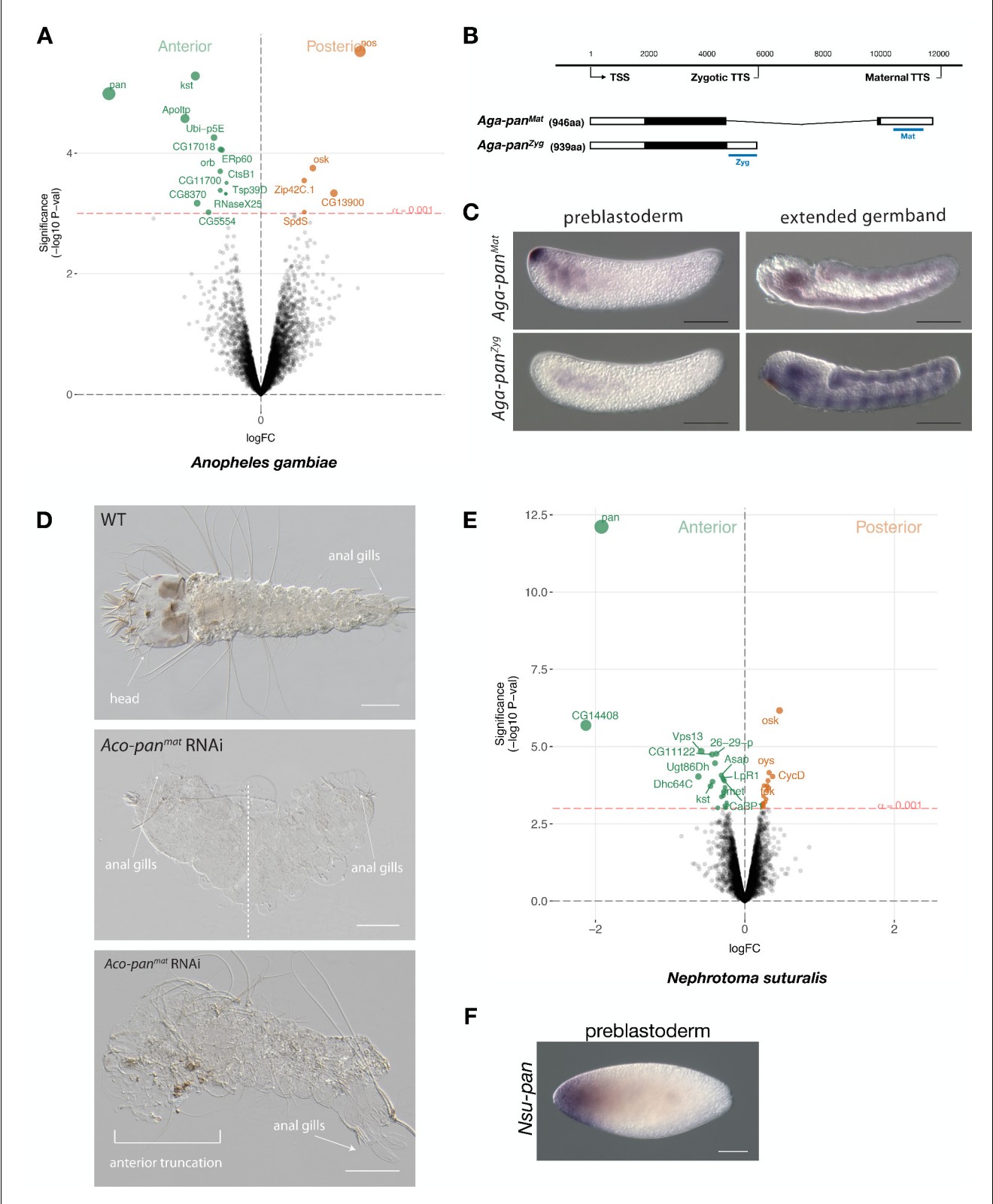

**Figure 6.** Expression and function of alternative *pangolin* isoforms in *Anopheles*. (**A**) Differential expression analysis of maternal transcripts between anterior and posterior halves of 1 hr-old *Anopheles* embryos. logFC: log fold-change. (**B**) Sketches of *Aga-pan^Mat* and *Aga-pan^Zyg* transcripts based on RNA-seq and RACE experiments (see also legend to *Figure 1A*). Blue lines the position of in situ hybridization probes and dsRNA. (**C**) RNA in situ hybridization of *Aga-pan^Mat* and *Aga-pan^Zyg* transcripts in 1 hr-old preblastoderm and germband extending embryos of *A.gambiae*. Anterior is left and

*Figure 6 continued on next page*

*Figure 6 continued*

dorsal up. Scale bar: 100μm. (**D**) 1 st instar *A.coluzzii* larval cuticle of wild type (top). Cuticle phenotypes of double abdomen (middle; 3/37) and intermediate, anterior truncation phenotype (bottom; 8/37) following *Aco-pan*$^{Mat}$ RNAi. Scale bar: 100μm. (**E**) Differential expression analysis of maternal transcripts between anterior and posterior halves of 1 hr-old *Nephrotoma* (Tipulidae) embryos. logFC: log fold-change. (**F**) RNA in situ hybridization of *Nsu-pan*$^{Mat}$.

DOI: https://doi.org/10.7554/eLife.46711.017

The following figure supplements are available for figure 6:

**Figure supplement 1.** Protein alignment of dipteran Pangolin orthologs and Panish.

DOI: https://doi.org/10.7554/eLife.46711.018

**Figure supplement 2.** RNA in situ hybridization of *Aco-pan*$^{Mat}$ and *Aco-pan*$^{Zyg}$ transcripts in 1 hr-old *A.coluzzii* preblastoderm embryos.

DOI: https://doi.org/10.7554/eLife.46711.019

**Figure supplement 3.** Stage-specific RNA-seq read coverage of *Ast-pan* genomic locus and sketches of *Ast-pan*$^{Mat}$ and *Ast-pan*$^{Zyg}$ transcripts.

DOI: https://doi.org/10.7554/eLife.46711.020

**Figure supplement 4.** Cuticle phenotype of a *A.coluzzii* 1 st instar larva following *Aco-pan*$^{Zyg}$ RNAi (11/23).

DOI: https://doi.org/10.7554/eLife.46711.021

**Figure supplement 5.** *panish* RNAi rescue experiments with *Cri-pan* mRNA.

DOI: https://doi.org/10.7554/eLife.46711.022

hr-old preblastoderm embryos and only weakly expressed in elongated germbands, whereas *Aga-pan*$^{Zyg}$ was expressed segmentally in elongated germbands but not in embryos younger than 2 hr-old preblastoderm embryos (***Figure 6C***). Both *pangolin* isoforms were conserved in *Anopheles coluzzii* with sequence identity above 99% and the maternal isoform (*Aco-pan*$^{Mat}$) was localized at the anterior pole (***Figure 6—figure supplement 2***). The stage-specific expression of both *pangolin* isoforms was also conserved outside the *Anopheles gambiae* species complex in *Anopheles stephensi* (***Figure 6—figure supplement 3***). Alignments of dipteran Pangolin proteins suggest that, in *Anopheles*, the maternal variant includes an additional seven amino acids at C-terminal end due to alternative polyadenylation and splicing (***Figure 6—figure supplement 1***).

To investigate the function of localized maternal *pangolin* expression in *Anopheles*, we specifically targeted this isoform in *Anopheles coluzzii*. Injection of *Aco-pan*$^{Mat}$-specific dsRNA into several hundred 1 hr-old *A. coluzzii* embryos resulted in only 37 cuticles with variable phenotypes, including anterior truncations and, in extreme cases, double abdomens (***Figure 6D***). We noticed perturbed segmentation boundaries in the double abdomens, suggesting that *Aga-pan*$^{Mat}$ may also function in segmentation, as suggested by its weak zygotic expression pattern. Injection of *Aco-pan*$^{Zyg}$-specific dsRNA into 1 hr-old *A. coluzzii* embryos resulted in severe segmentation defects that were difficult to characterize, but no double abdomens or anterior-specific truncation defects were found (***Figure 6—figure supplement 4***). Taken together with the isoform-specific transcript localization data presented above, these RNAi results support the hypothesis that *pangolin* acquired the anterior determinant function via the localization of a maternal transcript isoform with an alternative 3' end.

## Localization of maternal *pangolin* transcript in crane flies suggests that *pangolin* functioned as anterior determinant in ancestral flies

Anterior-localized maternal *pangolin* (*Tc-pan*) transcript has also been observed in the eggs of a beetle (*Tribolium castaneum*) (***Bucher et al., 2005***), but the function of this transcript remains unknown. Previous *Tc-pan* RNAi experiments targeted both maternal and zygotic transcripts and only revealed a function in posterior development, due to the role of zygotic *Tc-pan* in canonical Wnt signaling in the posterior growth zone (***Bolognesi et al., 2008***; ***Fu et al., 2012***; ***Prühs et al., 2017***; ***Ansari et al., 2018***). To test whether ancestral dipterans localized maternal *pangolin* transcript at the anterior pole of the egg, we established a culture of the crane fly *Nephrotoma suturalis* (Tipulidae), which belongs to the Tipulomorpha, one of the oldest branches of dipterans (***Grimaldi and Engel, 2005***; ***Wiegmann et al., 2011***). We sequenced the anterior and posterior transcriptomes of freshly deposited *Nephrotoma* egg halves and ranked 5371 transcripts according to the magnitude of their differential expression scores and P values (***Figure 6E***). The most enriched transcript in the posterior embryo was related to *oskar*, suggesting that crane fly eggs contain maternal germ plasm at the posterior pole. The most enriched transcript in the anterior embryo was homologous to *pangolin* and therefore named *Nsu-pan*. The anterior localization of this transcript was confirmed by RNA in

situ hybridization (*Figure 6F*). RACE experiments with cDNA from 1 hr-old embryos identified multiple isoforms with slightly variable 5' ends but the same open reading frame. An alignment of the predicted Nsu-Pan protein from this open reading frame with other dipteran Pangolin homologs revealed conserved N-terminal and C-terminal ends in Nsu-Pan. Taken together with our *Anopheles* data, our results in *Nephrotoma* suggest that ancestral dipteran insects localized maternal *pangolin* transcript in the anterior egg pole, where this transcript may have functioned as anterior determinant.

### Pangolin cannot substitute for Panish in Chironomnus

In the midge *Chironomus*, the ortholog of *pangolin* (*Cri-pan*) is not expressed maternally but its diverged paralog *panish* functions maternally as anterior determinant (*Klomp et al., 2015*). Given that *panish* evolved from *pangolin* via gene duplication, *panish* probably inherited its role from *pangolin*. Therefore, it is possible that *Cri-pan* and *panish* are still functionally equivalent when expressed at the anterior pole of preblastoderm *Chironomus* embryos. Alternatively, Panish may have co-evolved with the targets required for anterior patterning and Pangolin can no longer interact with those targets. In this case, *Cri-pan* should no longer be able to fulfill the function of *panish*. To distinguish between these possibilities, we examined the ability of *panish* and *Cri-pan* mRNAs to rescue the RNAi phenotype of *panish*. We have previously shown that dsRNA of the *panish* 3'UTR can induce the double abdomen phenotype with a penetrance of nearly 100%, and that this phenotype can be rescued in roughly half of the embryos by injecting *panish* mRNA with heterologous UTRs at the anterior pole, shortly after the injection of dsRNA (*Klomp et al., 2015*). We used this assay to compare the functions of *panish* mRNA (positive control), frame shifted *panish* mRNA (negative control), *Cri-pan* mRNA, and a modified *Cri-pan* mRNA designed to better resemble *panish* mRNA (*Cri-pan*^trunc. mRNA). *Cri-pan*^trunc. mRNA encodes a N-terminal truncated Cri-Pan variant, lacking the ß-Catenin binding and HMG box domains, with two mutations in the cysteine-clamp domain to mimic conserved changes of the Panish cysteine-clamp (*Figure 6—figure supplement 5A*). Only *panish* mRNA rescued *panish* RNAi embryos (*Figure 6—figure supplement 5B*), suggesting that *panish* co-evolved with its targets and functionally diverged after its origin via gene duplication from *pangolin*.

## Discussion

### Role of alternative transcription in the evolution of embryonic axis determinants from old genes

In this study, we have identified three unrelated old genes that encode the anterior determinant in moth flies, culicine mosquitoes, and anopheline mosquitoes, respectively (*Figure 7*). All three genes not only localize their maternal transcript at the anterior egg pole; they also are subject to alternative transcription, which allows a single gene to generate multiple transcript isoforms with distinct 5' and 3' ends through the use of alternative promoters (alternative transcription initiation) and polyadenylation signals (alternative transcription termination).

In moth flies, the localized maternal *odd-paired* transcript that functions as anterior determinant has an alternative first exon compared to the canonical isoform (*Figure 1B–D*, *Figure 2*, and *Figure 3A–B*). In the case of mosquitoes, maternal transcript isoforms of *cucoid* (in culicine mosquitoes) or *pangolin* (in anopheline mosquitoes) with alternative last exons are localized at the anterior pole of the egg and function as anterior determinant (*Figure 4*, *Figure 5*, and *Figure 6*). Since anterior determinants are localized in the anterior egg, and signals for the subcellular localization of transcripts are typically found in UTRs (*Holt and Bullock, 2009*), it is possible that alternative transcription facilitates the evolution of anterior determinants by providing the UTR sequence for isoform-specific localization signals that do not interfere with other gene functions. For example, it has been shown that alternative last exons of transcript isoforms confer isoform-specific localization in neurons (*Taliaferro et al., 2016*; *Ciolli Mattioli et al., 2019*). Additional experiments will be needed to test whether the unique UTR sequences of anterior determinants are essential for their localization at the anterior egg pole.

In addition to changes in UTR sequences, alternative transcription also can result in the truncation or elongation of the open reading frame. For example, the anterior determinant of *Clogmia* (Cal-

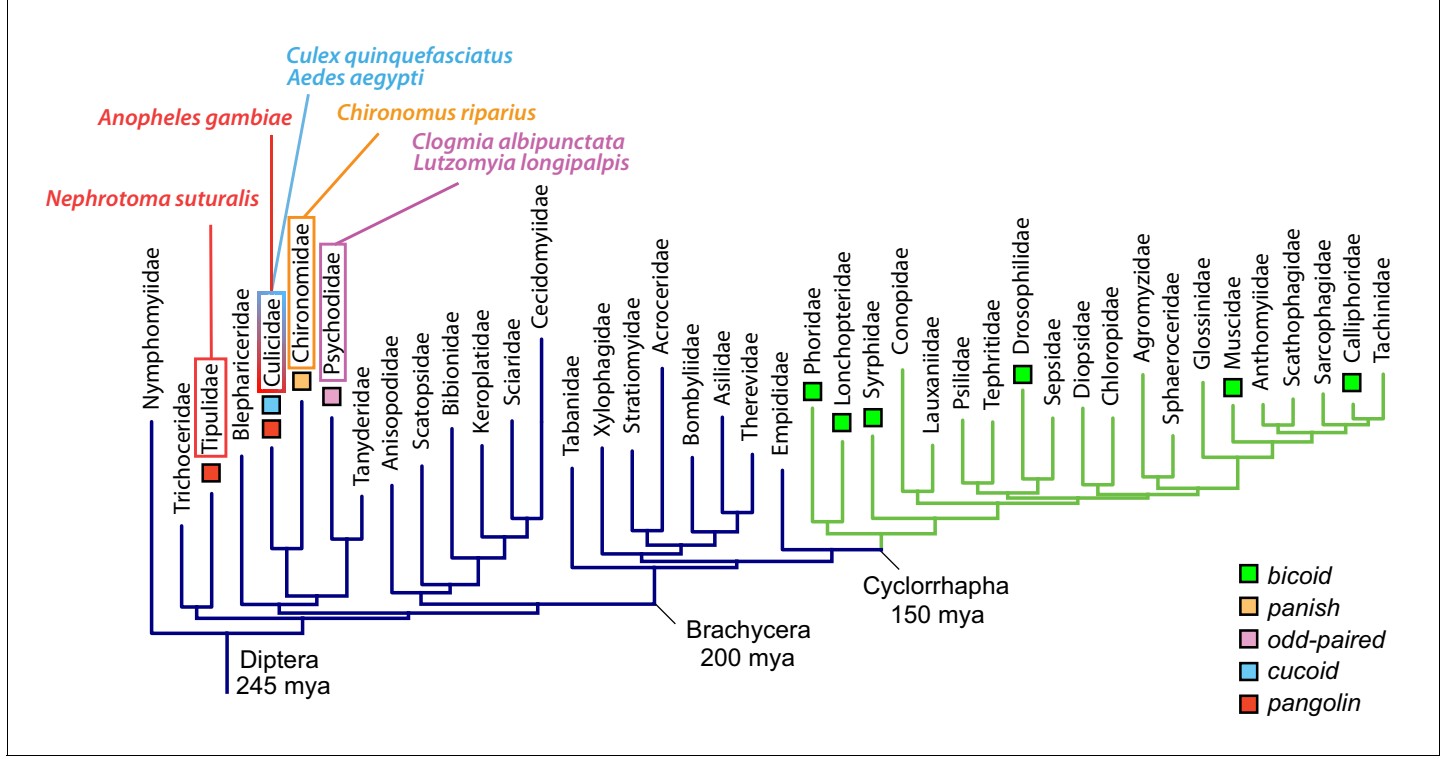

**Figure 7.** Anterior axis determinants in Diptera. The phylogenetic tree of dipteran families is based on published data and shows Cylorrhapha in green (**Wiegmann et al., 2011**). Mya, million years ago.

DOI: https://doi.org/10.7554/eLife.46711.023

The following figure supplements are available for figure 7:

**Figure supplement 1.** Occurrence of *panish* in Chironomidae genomes.

DOI: https://doi.org/10.7554/eLife.46711.024

**Figure supplement 2.** Maternal and zygotic transcript variant of *Clogmia zerknüllt* (*Cal-zen*).

DOI: https://doi.org/10.7554/eLife.46711.025

Opa$^{Mat}$) lacks the N-terminal 20 amino acids (**Figure 1—figure supplement 1**), and the anterior determinant of *Anopheles* (Aga-pan$^{Mat}$) encodes protein that includes additional seven amino acids at the C-terminal end (**Figure 6—figure supplement 1**). However, these changes to the protein may not have been important for adopting a function as anterior determinant. The truncation in the maternal Odd-paired protein that we observed in *Clogmia* is not conserved in *Lutzomyia*, in which *Llo-opa$^{Mat}$* and *Llo-opa$^{Zyg}$* encode the same protein, and full-length Odd-paired homologs from these and other species can function as anterior determinant in *Clogmia* (**Figure 3B–C**). Also, the elongation of Aga-Pan$^{Mat}$ protein is not conserved in *Nephrotoma*, in which the localized *Nsu-pan$^{Mat}$* transcript encodes a Pangolin protein with a conserved C-terminal end (**Figure 6—figure supplement 1**). Therefore, modifications in the open reading frame of these genes may reflect secondary changes.

## Evolution of new genes that encode embryonic axis determinants

Unlike the anterior determinants identified in this study, the previously described anterior determinants of *Drosophila* and *Chironomus* are encoded by newly evolved genes, *bicoid* and *panish*. These genes seem to be dispensable outside the context of axis specification (**Driever, 1993**; **Klomp et al., 2015**), suggesting that they evolved specifically for this function. They could have acquired their function de novo via protein evolution or via inheritance from the progenitor gene. Our findings suggest that the role of *pangolin* in axis specification was already present in ancestral dipterans (**Figure 7**). We therefore propose that *panish,* which evolved from *pangolin* via gene duplication in the Chironominae lineage (**Figure 7—figure supplement 1**), inherited its function from

*pangolin*. Future examinations of *pangolin* isoforms and their expression in the eggs of chironomids that lack *panish* orthologs (species representing basal chironomid lineages) could reveal intermediate steps in this process, such as a localized truncated *pangolin* isoform.

Similarly, *bicoid* could have acquired its function de novo via protein evolution or via inheritance from its progenitor gene, *zerknüllt*. Several previous studies have hypothesized that Bicoid replaced Orthodenticle, a conserved homeodomain protein with similar DNA-binding affinity that functions in animal head development (*Wimmer et al., 2000*; *Schröder, 2003*; *Lynch et al., 2006*; *Datta et al., 2018*). Ancestrally reconstructed homeodomains confirmed that a single amino acid change in the homeodomain of Bicoid (Q50K), which is shared by Bicoid and Orthodenticle, caused a dramatic shift of Bicoid's DNA-binding affinity in vitro and target recognition in vivo (*Liu et al., 2018*).

We cannot rule out that *orthodenticle* functioned as anterior determinant in ancestral brachyceran flies. However, in analogy with our findings, it is also possible that a maternal *zerknüllt* isoform became localized at the anterior pole of the egg and acquired the role of anterior determinant via co-option, prior to the origin of *bicoid* via gene duplication (*Figure 7—figure supplement 2*). Maternal *zerknüllt* expression is common in lower Diptera but was lost in Cyclorrhapha (*Stauber et al., 2002*). If *bicoid* inherited its function from *zerknüllt*, the Q50K mutation in the homeodomain of Bicoid must have been a secondary, potentially maladaptive change. In this case, it may have been fixed in the cyclorrhaphan stem lineage via a compensatory or balancing mechanism and would have driven co-evolution of its targets.

It may be objected that, in *Drosophila*, reverting the K50 residue of the Bicoid homeodomain to Q50 is lethal and results in a *bicoid* null phenotype (*Liu et al., 2018*). However, how the ancestral gene network responded to the Q50K mutation of Bicoid cannot be inferred from observations in *Drosophila*. Moreover, published biochemical data suggest that the Q50K mutation increases interaction with the consensus Bicoid binding DNA motif much stronger than it reduces interaction with the consensus Zerknüllt binding DNA motif. It is therefore conceivable that the Q50K mutation had a less dramatic effect in ancestral flies, in which the target genes of the anterior determinant were activated via Zerknüllt binding sites, than in *Drosophila*, in which the target genes of the anterior determinant are activated via Bicoid binding sites. Examination of the mechanisms that determine embryo polarity in non-cyclorrhaphan Brachycera flies might help to test this hypothesis. If *panish* and *bicoid* inherited their functions, their evolutionary origin and divergence could have served the purpose of reducing the pleiotropy of their progenitor genes, *pangolin* and *zerknüllt*, rather than allowing them to take on an entirely new function in development.

## Role of alternative transcription in the evolution

Recent genome-wide analyses have shown that alternative transcription is a widespread phenomenon. For example, there are on average four alternative transcription start sites per gene in humans (*Forrest et al., 2014*) and at least 50–70% of mammalian genes are subject to alternative polyadenylation (*Shepard et al., 2011*; *Adams et al., 2000*; *Derti et al., 2012*). Alternative transcript isoforms can be tightly regulated in a cell or tissue specific manner and can affect transcription and translation efficiency as well as splicing (*An et al., 2008*; *Davuluri et al., 2008*; *Lau et al., 2010*; *Pinto et al., 2011*; *Smibert et al., 2012*; *Anvar et al., 2018*; *Tushev et al., 2018*; *Taliaferro et al., 2016*; *Ciolli Mattioli et al., 2019*; *Shabalina et al., 2010*). Functional studies in model organisms have shown that alternative transcription can generate dominant negative and alternatively localized protein isoforms (*Davuluri et al., 2008*; *Bharti et al., 2008*; *Vacik et al., 2011*; *Berkovits and Mayr, 2015*; *Schrankel et al., 2016*), while misregulation of alternative transcript isoforms has been associated with human diseases including cancer (*Mayr and Bartel, 2009*; *Wiesner et al., 2015*; *Pal et al., 2012*; *Shapiro et al., 2011*). However, the contribution of alternative promoters (alternative transcription initiation) and polyadenylation signals (alternative transcription termination) to the evolution of new gene functions and regulatory networks remains poorly understood (*Carroll et al., 2005*; *Peter and Davidson, 2011*; *Wittkopp and Kalay, 2011*; *de Klerk and 't Hoen, 2015*).

The results of several previous studies suggest that alternative transcription may underlie the evolutionary diversification of gene functions. For example, a large fraction of alternative promoter sequences is conserved between human and mice, but those with cell or tissue restricted expression have frequently changed during mammalian evolution (*Baek et al., 2007*; *Forrest et al., 2014*), suggesting that alternative promoters may have played a significant role in cell type evolution. Another likely substrate for evolutionary diversification are the protein terminal ends generated by alternative

transcription in conjunction with alternative splicing. These terminal ends commonly contain intrinsically disordered regions which are enriched in sites that mediate protein-protein interactions (*Buljan et al., 2013*; *Shabalina et al., 2014*). A recent case study found that a light fur color variant of beach mice evolved repeatedly via selection for an alternative *agouti* isoform with increased translation efficiency (*Linnen et al., 2013*; *Mallarino et al., 2017*).

Our in vivo study revealed three old genes that evolved the anterior determinant function by localizing an alternative transcript isoform at the anterior pole of the egg. Therefore, we propose that differential expression of alternative transcript isoforms can result in the evolution of new gene functions, independent of, and prior to gene duplication and sub-functionalization. Given that alternative transcription is a widespread phenomenon, it could play an important role in the evolution of gene regulatory networks.

# Materials and methods

### Key resources table

| Reagent type (species) or resource | Designation | Source or reference | Identifiers | Additional information |
|---|---|---|---|---|
| Strain, strain background (*Nephrotoma suturalis*) | Nsu | https://doi.org/ 10.1083/jcb.25.1.95 | NA | |
| Strain, strain background (*Chironomus riparius*) | Cri | https://doi.org/ 10.1126/science.aaa7105; https://doi.org/ 10.1111/mec.1411 | CRIP_Laufer | |
| Strain, strain background (*Clogmia albipunctata*) | Cal | https://doi.org/ 10.1007/s004270050238 | NA | |
| Strain, strain background (*Culex quinquefasciatus*) | Cqu | NIAID | NA | |
| Strain, strain background (*Anopheles gambiae*) | A.gambiae | NIAID | G-3 strain | |
| Strain, strain background (*Anopheles coluzzii*) | A.coluzzii | Insect transformation facility, University of Maryland | NA | |
| Strain, strain background (*Aedes aegypti*) | Aae | NIAID | Liverpool 'Black eye' | |
| Gene (*Clogmia albipunctata*) | *Cal-opa$^{Mat}$* | This paper | MN122104 | See Materials and methods - Cloning procedures and mRNA/dsRNA synthesis; RNA in situ hybridization; Rapid Amplification of cDNA Ends (RACE) |
| Gene (*Clogmia albipunctata*) | *Cal-opa$^{Zyg}$* | This paper | MN122105 | See Materials and methods - Cloning procedures and mRNA/dsRNA synthesis; RNA in situ hybridization; Rapid Amplification of cDNA Ends (RACE) |

*Continued on next page*

*Continued*

| Reagent type (species) or resource | Designation | Source or reference | Identifiers | Additional information |
|---|---|---|---|---|
| Gene (*Clogmia albipunctata*) | *Cal-slp* | This paper | MN122106 | See Materials and methods - Cloning procedures and mRNA/dsRNA synthesis; RNA in situ hybridization |
| Gene (*Clogmia albipunctata*) | *Cal-mira* | This paper | MN122107 | See Materials and methods - Cloning procedures and mRNA/dsRNA synthesis; RNA in situ hybridization |
| Gene (*Clogmia albipunctata*) | *Cal-otd* | This paper | MN122108 | See Materials and methods - RNA in situ hybridization |
| Gene (*Clogmia albipunctata*) | *Cal-nos1* | This paper | MN122109 | See Materials and methods - RNA in situ hybridization |
| Gene (*Clogmia albipunctata*) | *Cal-nos2* | This paper | MN122110 | See Materials and methods - RNA in situ hybridization |
| Gene (*Clogmia albipunctata*) | *Cal-nos3* | This paper | MN122111 | See Materials and methods - RNA in situ hybridization |
| Gene (*Clogmia albipunctata*) | *Cal-nos4* | This paper | MN122112 | See Materials and methods - RNA in situ hybridization |
| Gene (*Clogmia albipunctata*) | *Cal-vas* | This paper | MN122113 | See Materials and methods - RNA in situ hybridization |
| Gene (*Clogmia albipunctata*) | *Cal-tud* | This paper | MN122114 | See Materials and methods - RNA in situ hybridization |
| Gene (*Clogmia albipunctata*) | *Cal-gcl* | This paper | MN122115 | See Materials and methods - RNA in situ hybridization |
| Gene (*Lutzomyia longipalpis*) | *Llo-opa*[Mat] | This paper | MN122116 | See Materials and methods - Rapid Amplification of cDNA Ends (RACE) |
| Gene (*Lutzomyia longipalpis*) | *Llo-opa*[Zyg] | This paper | MN122117 | See Materials and methods - Rapid Amplification of cDNA Ends (RACE) |
| Gene (*Lutzomyia longipalpis*) | *Llo-osk* | This paper | MN122118 | See Materials and methods - RNA in situ hybridization |
| Gene (*Chironomus riparius*) | *Cri-opa* | This paper | MN122119 | See Materials and methods - RNA in situ hybridization |
| Gene (*Culex quinquefasciatus*) | *cucoid*[A] | This paper | MN122120 | See Materials and methods - RNA in situ hybridization |
| Gene (*Culex quinquefasciatus*) | *cucoid*[B] | This paper | MN122121 | See Materials and methods - RNA in situ hybridization |

*Continued*

| Reagent type (species) or resource | Designation | Source or reference | Identifiers | Additional information |
|---|---|---|---|---|
| Gene (*Culex quinquefasciatus*) | *cucoid$^C$* | This paper | MN122122 | See Materials and methods - Cloning procedures and mRNA/dsRNA synthesis; RNA in situ hybridization |
| Gene (*Culex quinquefasciatus*) | *Cqu-nos* | This paper | MN122123 | See Materials and methods - RNA in situ hybridization |
| Gene (*Aedes aegypti*) | *Aae-cucoid$^A$* | This paper | MN122124 | See Materials and methods - Rapid Amplification of cDNA Ends (RACE) |
| Gene (*Aedes aegypti*) | *Aae-cucoid$^B$* | This paper | MN122125 | See Materials and methods - Rapid Amplification of cDNA Ends (RACE) |
| Gene (*Aedes aegypti*) | *Aae-cucoid$^C$* | vectorbase.org | AAEL013321 | See Materials and methods - Rapid Amplification o f cDNA Ends (RACE) |
| Gene (*Aedes aegypti*) | *Aae-nos* | vectorbase.org | AAEL012107 | See Materials and methods - RNA in situ hybridization |
| Gene (*Anopheles gambiae*) | *Aga-pan$^{Mat}$* | This paper | MN122126 | See Materials and methods - Cloning procedures and mRNA/dsRNA synthesis; RNA in situ hybridization; Rapid Amplification of cDNA Ends (RACE) |
| Gene (*Anopheles gambiae*) | *Aga-pan$^{Zyg}$* | This paper | MN122127 | See Materials and methods - Cloning procedures and mRNA/dsRNA synthesis; RNA in situ hybridization; Rapid Amplification of cDNA Ends (RACE) |
| Gene (*Anopheles gambiae*) | *Aga-cad* | This paper | MN122128 | See Materials and methods - RNA in situ hybridization |
| Gene (*Nephrotoma suturalis*) | *Nsu-pan* | This paper | MN122129 | See Materials and methods - RNA in situ hybridization |
| Sequence-based reagent | dsRNAs | This paper | | see Materials and methods - Cloning procedures and mRNA/dsRNA |
| Sequence-based reagent | RNA probes | This paper | | See Materials and methods - RNA in situ hybridization |
| Commercial assay or kit | SMARTer RACE 5'/3' Kit | Takara | 634858 | |
| Commercial assay or kit | mMESSAGE mMACHINE SP6 | Thermo Fisher | AM1340 | |

*Continued on next page*

*Continued*

| Reagent type (species) or resource | Designation | Source or reference | Identifiers | Additional information |
|---|---|---|---|---|
| Commercial assay or kit | QuikChange Lightning Site-Directed Mutagenesis Kit | Agilent | 210515 | |
| Software, algorithm | Geneious | https://www.geneious.com | RRID: SCR_010519 | version 11.1.5 |
| Software, algorithm | GraphPad Prism | https://graphpad.com | RRID: SCR_015807 | version 1.4 |

## Cloning procedures and mRNA/dsRNA synthesis

Coding sequences from *Clogmia*, *Lutzomyia*, *Chironomus*, *Anopheles*, *Nephrotoma*, *Culex*, and *Aedes* were amplified from embryonic cDNA with primers constructed from RNA-seq data. Coding sequence of *odd-paired* was amplified from cDNA (FI01113) that was obtained from the BDGP Gold collection of the Drosophila Genomics Resource Center. Amplified cDNA was cloned into the expression vector pSP35T (*Amaya et al., 1991*), using In-Fusion HD Cloning Kit (Clontech), and PstI- or EcoRI-linearized vector was used for mRNA synthesis using mMESSAGE mMACHINE SP6. *panish*, *Cri-pan*, and *Cri-pan^trunc^* mRNAs were synthesized from PCR template (containing T7 polymerase binding site) using mMESSAGE mMACHINE T7ultra. Mutations in the open reading frame (*Cal-opa-^Zyg-Met21Leu^*, *panish FS*, *Cri-pan^trunc^*) and in the kozak sequences of *Dme-opa* and *Cri-opa* were generated using QuikChange Lightning Site-Directed Mutagenesis Kit (Agilent). Double-stranded RNA (dsRNA) was generated from PCR-amplified templates using embryonic cDNA and primers containing T7 polymerase binding sites as described (*Klomp et al., 2015*).

Forward and reverse primer sequences for generating templates for mRNA synthesis were:

Cal-opa^Mat^
5'-TAAGATGAGTCCGAATCACTTACTGGCC
5'-TTAATAGGCCGTCGCTGCACC
Cal-opa^zyg^
5'-CAACATGATGATGAACGCTTTTATGGAA
5'-TTAATAGGCCGTCGCTGCACC
Cal-opa^115-655^
5'-ATGCTCTTCTCAAATCACTCTTCAGC
5'-TTAATAGGCCGTCGCTGCACC
Cal-opa^182-655^
5'-ATGAACCCGGGAACCTTGGG
5'-TTAATAGGCCGTCGCTGCACC
Cal-opa^182-445^
5'-ATGAACCCGGGAACCTTGGG
5'-TTACGCGGGATTCAGCTGACTATG
Cal-opa^1-337+22^
5'-CAACATGATGATGAACGCTTTTATGGAA
5'-TCACAAAATTTCACTGAATTCCGTCAAAATATCACTAGA
Llo-opa
5'-AAAGATGATGATGAATGCATTTATGGACACAG
5'-TCAGTACGCCGTGGCGGCG
Dme-opa^Dme kozak^
5'-GACCATGATGATGAACGCCTTCA
5'-GTCAATACGCCGTCGCTGCGCCGGG
Cri-opa^Cri kozak^
5'-AAAAATGATGATGAATGGTTTTATGGACACA
5'-TCAATAAGCTGTCGTTGGACCGTGAT
Cri-pan^trunc^
5'- AAAAATGTATCCAGATTGGAGCTCGC
5'- TTACGTCACACTAATAGCATTTCCATCATCCC

Forward and reverse primer sequences for dsRNA (lengths of dsRNAs in brackets; gene specific sequence of primers underlined):

*Cal-opa^Mat* (222 bp):
5'-CAGAGATGCATAATACGACTCACTATAGGGAGA<u>AAACAATTGTGAAGTGCGACA</u>
5'-CAGAGATGCATAATACGACTCACTATAGGGAGA<u>CAAATTTCCAAACGATGACAGA</u>
*Cal-opa^Zyg* (315 bp):
5'- CAGAGATGCATAATACGACTCACTATAGGGAGA<u>ACTACCGCCGCGAACACACG</u>
5'-CAGAGATGCATAATACGACTCACTATAGGGAGA<u>GTCCAGTCGATTCCATAAAAG</u>C
*Cal-slp* (927 bp):
5'- CAGAGATGCATAATACGACTCACTATAGGGAGA<u>TCGATCAGCTCCCTTTTGCC</u>
5'-CAGAGATGCATAATACGACTCACTATAGGGAGA<u>TGAGATCGTTCCCGTTGGAC</u>
*Cal-mira* (993 bp):
5'- CAGAGATGCATAATACGACTCACTATAGGGAGA<u>ACAGCAAAAAGGAAGCGAAA</u>
5'-CAGAGATGCATAATACGACTCACTATAGGGAGA<u>GGGATTCAATTTGCCTTTGA</u>
*Aga-pan^Mat* (976 bp):
5'- CAGAGATGCATAATACGACTCACTATAGGGAGA<u>CACACAGGGCACAATAATCG</u>
5'- CAGAGATGCATAATACGACTCACTATAGGGAGA<u>GACTGCATGTCCGTCGTCTA</u>
*Aga-pan^Zyg* (843 bp):
5'- CAGAGATGCATAATACGACTCACTATAGGGAGA<u>ACATCACACACCCCACACAC</u>
5'- CAGAGATGCATAATACGACTCACTATAGGGAGA<u>TTGGTCCGTTCGTGATTGTA</u>
*cucoid* (926 bp):
5' - CAGAGATGCATAATACGACTCACTATAGGGAGA<u>CGAGGATGTTGCTGGAGAAT</u>
5'- CAGAGATGCATAATACGACTCACTATAGGGAGA<u>ACTCCCGAAATCGGAAAACT</u>
*Aae-cucoid* (857 bp):
5'-CAGAGATGCATAATACGACTCACTATAGGGAGA<u>CGACAAGCCCTACAAATGCT</u>
5'-CAGAGATGCATAATACGACTCACTATAGGGAGA<u>TGATCTGGATGTTGCCGTAG</u>

## Microinjection of embryos

*Chironomus* embryo injection was done as previously described (*Klomp et al., 2015*). *Clogmia* eggs were dissected from ovaries and activated under water. Eggs of *Aedes*, *Culex*, and *Anopheles* were collected in a dark chamber on a moist filter paper for about 30 min (Fisher Scientific, Cat. No 09–795C). These eggs were transferred to another filter paper cut into 4 cm x 2 cm pieces and aligned perpendicularly to the edge of a cover glass (Fisher Scientific, Cat. No 12-648-5C) with the prospective injection side pointing towards the glass edge. We noticed that injecting eggs near the anterior or posterior pole was critical for survival of the procedure. During the alignment procedure, water was applied to the filter paper as needed to prevent eggs from desiccation. After aligning the eggs, the cover glass was removed and excess water on the filter paper was absorbed using filter paper. A second cover glass with a layer of double-sided tape (Scotch 3M) was slightly pressed against the aligned eggs to transfer the eggs to the double-sided tape. The embryos were then immediately covered with halocarbon oil to prevent desiccation. For cuticle preparations, the embryos were injected under halocarbon oil 27 (Sigma, MKBZ7202V). The oil was washed off under a gentle stream of water immediately after injection. In the case of *Clogmia*, *Aedes*, and *Culex* the cover glass was transferred to a moist chamber (petri dish with wet kimwipe paper) and kept at 28 ˚C, and water was added every day to prevent desiccation. In the case of *Anopheles*, the eggs were allowed to develop under water. Removal of the halocarbon oil was critical to ensure embryo survival until late developmental stages and hatching. For eggs to be fixed within a day following injection, we used a 1:1 mixture of halocarbon oil 27 and halocarbon oil 700 (Sigma, MKCB5817) and left the injected eggs immersed in the oil until fixation. Embryos were injected with quartz needles using a Narishige IM-300 microinjector. Quartz capillaries (Sutter Instruments Q100-70-10) were pulled with a Sutter instrument P-2000 laser-based micropipette puller. Our settings for the needle puller were: Heat 645, Fil 4, Vel 40, Del 125, Pul 130. Needles were back-filled and the tip was broken open at the time of injection by slightly touching the first egg.

## Embryo fixation

*Clogmia* Embryos were dechorionated using a 10% dilution of commercial bleach (8.25% sodium hypochloride) for 3 min. For *Nephrotoma* embryos, a 25% dilution was used for 3 min until the

chorion became slightly transparent. Embryos of *Aedes*, *Culex*, and *Anopheles* were dechorionated as described (*Juhn and James, 2012*). Dechorionated embryos were fixed in a 50 mL falcon tube, using 20 mL of boiling salt/detergent-solution (100 µL 10% triton-X, 500 µL 28% NaCl, up to 20 mL of water). After 10 s, water was applied to the tube to cool down the embryos. If needed, the embryos were devitellinized in a 1:1 mixture of n-heptane and methanol by gentle shaking. Embryos with vitelline membrane attached were further devitellinized using sharp tungsten needles in an agar plate covered with methanol. Devitellized embryos were stored in 100% methanol at −20℃.

## RNA in situ hybridization

RNA in situ hybridizations were conducted as described (*Klomp et al., 2015*), using digoxigenin (DIG)-labeled probes and Fab fragments from anti-DIG antibodies conjugated with alkaline phosphatase (AP) (Roche, IN, USA). Probes were prepared from PCR templates, using sequence-specific forward primers and reverse primers with T7 promoter sequence (see above for *Cal-opa^Mat*, *Cal-opa^Zyg*, *Cal-cad*, *Cal-slp*, *Cal-mira*, *Aga-pan^Mat*, *Aga-pan^Zyg*, and *Aae-cucoid*; gene specific sequence underlined).

*Cal-nos1* (450 bp):
5'-AGCACTTTTCCCCCAAGAGT
5'-CAGAGATGCATAATACGACTCACTATAGGGAGAGGCATTCATATTTCCTCAGCA

*Cal-nos2* (475 bp):
5'-AATTATTCTGTTCCAAAGTTGAGATT
5'-CAGAGATGCATAATACGACTCACTATAGGGAGACCCCAGACTGGTGACAAAT

*Cal-nos3* (548 bp):
5'-TGAGTTAAATAGAGTGAAAACAGCAAA
5'-CAGAGATGCATAATACGACTCACTATAGGGAGATACCGTCTCGTGCTTAATCG

*Cal-nos4* (440 bp):
5'-GGCAAAATTTTCCAAGTGAA
5'-CAGAGATGCATAATACGACTCACTATAGGGAGACGTGTCCTCAAGCGTGTAGAT

*Cal-vas* (938 bp):
5'-CTGAGGCGAACTTGTGTGAA
5'-CAGAGATGCATAATACGACTCACTATAGGGAGAATTGGCAATGTCCAGTCCTC

*Cal-tud* (921 bp):
5'-ATTCTGCAAGTCGTCGAGGT
5'-CAGAGATGCATAATACGACTCACTATAGGGAGACCTGTACCAGCCATTGTCCT

*Cal-gcl* (450 bp):
5'-GCAGAACCCCTTGGACATTA
5'-CAGAGATGCATAATACGACTCACTATAGGGAGAGTAACGCCCACAATTCGTCT

*cucoid^A* (939 bp):
5'-ACGATGAGGAGGAGGGTTCT
5'- CAGAGATGCATAATACGACTCACTATAGGGAGACGCACTTCACCGTGTGTAAC

*cucoid^B* (717 bp):
5'-GGGGCGACATCTATATCTCACT
5'-CAGAGATGCATAATACGACTCACTATAGGGAGAACAGTGAGAAAAATTCCCAACTTTAGT

*cucoid^C* (926 bp):
5'-CGAGGATGTTGCTGGAGAAT
5'-CAGAGATGCATAATACGACTCACTATAGGGAGAACTCCCGAAATCGGAAAACT

*Cqu-cad* (956 bp):
5'-CACGTGTTCCATCAGTCCAG
5'-CAGAGATGCATAATACGACTCACTATAGGGAGAATGAGGCTTAACGAGGATGG

*Cqu-nos* (927 bp):
5'-AAGTGCCGTGAATTTTGTCC
5'-CAGAGATGCATAATACGACTCACTATAGGGAGAGCGAAACCAATTCGACAGTT

*Nsu-pan* (966 bp):
5'-TCGCGGCAAGATCATAGTCC
5'-CAGAGATGCATAATACGACTCACTATAGGGAGACCTGCAGGGTTTACACCACT

*Aae-nos* (914 bp):
5'- CAAACGTGAAGCGGAAGATT
5'- CAGAGATGCATAATACGACTCACTATAGGGAGAATTACGTCCGGAAGTGTTCG

## Rapid Amplification of cDNA Ends (RACE)

Total RNA was phenol/chloroform extracted from *Clogmia* (1 hr-old and 9 hr-old embryos), *Anopheles* (1–6 hr-old embryos), *Culex* (0–7 hr old embryos), and *Nephrotoma* (1–29 hr-old embryos) fixed in TRIzol Reagent (Invitrogen) and precipitated with isopropanol. 5'/3' RACE was performed using SMARTer RACE 5'/3' Kit (Clontech) with the custom-made primers (including at the 5' end 15 nucleotides of pRACE vector sequence). Gene specific sequences are underlined.

*Cal-opa* 5'RACE primer: 5'-GATTACGCCAAGCTTCTGGGTGACGCCGTGGGCAAGGACGTCA
*Cal-opa* 3'RACE primer: 5'-GATTACGCCAAGCTTCGCGTCGATCGTCACGCCCCCAAATTCG
*Aga-pan* 5'RACE primer: 5'-GATTACGCCAAGCTTCGAATCTCCGGCCGCGGAATTGAGACTT
*Aga-pan* 3'RACE primer: 5'-GATTACGCCAAGCTTAGCTTCACGCGACCAGCAAAACCAACGG
*cucoid* 5'RACE primer: 5'- GATTACGCCAAGCTTCGTGACGGCTTCGATGGTTGGTTTTTCC
*cucoid* 3'RACE primer: 5'- GATTACGCCAAGCTTCGCACGTGTTGAACAGTCACATGTTGAC
*Aae-cucoid* 5'RACE primer: 5'- GATTACGCCAAGCTTGATCCGGTGGATCGGACTTGGCCGAGAT
*Aae-cucoid* 3'RACE primer: 5'- GATTACGCCAAGCTTAACCTCCCTCGGGGTTGAACGTGAAGCT
*Aae-cucoid* 5'RACE primer: 5'- GATTACGCCAAGCTTGATCCGGTGGATCGGACTTGGCCGAGAT
*Aae-cucoid* 3'RACE primer: 5'- GATTACGCCAAGCTTAACCTCCCTCGGGGTTGAACGTGAAGCT
*Nsu-pan* 5'RACE primer: 5'-GATTACGCCAAGCTTTCTGGTCGTGCGACGTTCTTCCAAATCG
*Nsu-pan* 3'RACE primer: 5'-GATTACGCCAAGCTTTCCCGTTGGTGCAAATCCACGAGATGTG

## Cuticle preparations

Cuticles were prepped four to five days after injection. Eggshells was removed with tungsten needles and the embryos were transferred to a glass block dish with a drop of 1:4 glycerol/acetic acid. Following incubation in 1:4 glycerol/acetic acid overnight at room temperature, the cuticles were transferred onto a glass slide, oriented, mounted in 1:1 Hoyer's medium/lactic acid (*Stern and Sucena, 2000*), covered with a cover glass, and dried overnight at 65 °C.

## RNA-seq sample preparation and sequencing

Bisection of anterior and posterior embryo halves, RNA extraction, and sequencing were conducted as described (*Klomp et al., 2015*). In the case of *Clogmia*, anterior or posterior embryo halves from three 1 hr-old embryos were pooled and RNA-seq data were obtained from two replicates. In case of *Lutzomyia*, embryo halves from ten 1–2 hr-old embryos were pooled four replicates were generated. In case of *Anopheles* (G-3 strain), embryo halves from five 1 hr-old embryos were pooled and three replicates were generated, In the case of *Culex*, embryo halves from seven 1 hr-old embryos were pooled and three replicates were generated. In case of *Aedes* (Liverpool 'black eye' strain), embryo halves from five 1 hr-old embryos were pooled and four replicates were generated. In case of *Nephrotoma*, embryo halves from nine 1 hr-old embryos were pooled and three replicates were generated. Stage-specific *Clogmia* transcriptomes were generated from the offspring of a single mother and total RNA from five embryos was used for each stage. In the case of *Lutzomyia*, about 100 staged embryos were pooled for RNA extraction, and two independent RNA extractions from each time point were combined and submitted for sequencing.

Prior to library construction, RNA integrity, purity, and concentration were assessed using an Agilent 2100 Bioanalyzer with an RNA 6000 Nano Chip (Agilent Technologies, USA). Purification of messenger RNA (mRNA) was performed using the oligo-dT beads provided in the Illumina TruSEQ mRNA RNA-SEQ kit (Illumina, USA). Complementary DNA (cDNA) libraries for Illumina sequencing were constructed using the Illumina TruSEQ mRNA RNA-SEQ kit (Illumina, USA), using the manufacturer-specified protocol. Briefly, the mRNA was chemically fragmented and primed with random oligos for first strand cDNA synthesis. Second strand cDNA synthesis was then carried out with dUTPs to preserve strand orientation information. The double-stranded cDNA was then purified, end repaired, and 'a-tailed' for adaptor ligation. Following ligation, the samples were selected a final library size (adapters included) of 400–550 bp using sequential AMPure XP bead isolation (Beckman Coulter, USA). The libraries were sequenced in an Illumina HiSeq 4000 DNA sequencer, utilizing a pair end sequencing flow cell with a HiSeq Reagent Kit v4 (Illumina, USA).

## RNA-seq data preprocessing

The TrimGalore (*Krueger, 2012*) wrapper for Cutadapt (*Martin, 2011*) and FastQC (*Andrews, 2010*) was used to remove adapters and low quality sequences from raw fastq files. Overlapping reads were combined with Flash (*Magoč and Salzberg, 2011*) prior to assembly.

## Transcriptome assembly and annotation

Trinity 2.4.0 (*Grabherr et al., 2011*) on the Indiana University Karst high-performance computing cluster was used for assembling contiguous sequences (contigs) from the paired end (PE) sequence data of *Clogmia*, *Lutzomyia*, *Anopheles*, and *Nephrotoma*. ABySS 2.0 (*Jackman et al., 2017*) was used for assembling contigs from *Culex* and *Aedes* data. Only contigs of 200 nucleotides or greater were retained. BLAST+ tools (*Camacho et al., 2009*) were used to annotate contigs by conducting best-reciprocal-blast first against the *Drosophila melanogaster* transcriptome (BDGP6) peptide sequences (blastx/tblastn) and then the coding sequence (tblastx) with a maximum threshold evalue of 1e-10. Biomart and AnnotationDbi packages were used for gene ids and names. The longest open reading frames (ORFs) of unannotated transcripts were compared to the RefSeq invertebrate protein database (downloaded 4-1-2017) using blastp (max evalue 1e-10) followed by a similar comparison to remove transcripts with ORFs matching RefSeq plant, protozoan, archaea, bacteria, fungi, plasmid, or viral sequences (downloaded 6-1-2017). Remaining transcripts were designated by the top BLAST hit in *D. melanogaster*.

## Alignment and differential expression analysis

Cleaned paired-end read data was aligned and analyzed using R base (*Ihaka and Gentleman, 1996*) and Bioconductor (*Gentleman et al., 2004*) software packages. Sequence alignment was conducted with the seed-and-vote aligner, Subread, as implemented in the Rsubread package (*Liao et al., 2013*) with up to five multi-mapping locations, six mismatches, and 20 subreads/seeds per read. Sequence file manipulation, including sorting and indexing of '.bam' files, was done using Rsamtools (*Morgan et al., 2013*).

To avoid potential biases in transcript localization unrelated to anterior-posterior axis formation, transcripts annotated with mitochondrial, ribosomal, or ambiguous status (e.g., predicted, hypothetical, or uncharacterized) were filtered out prior to the differential expression comparisons. Transcripts with 20 or fewer counts in any of the A-P pairs were also excluded from the analysis prior to library normalization. Lower scoring, potentially related transcripts matching a given gene from the *D. melanogaster* transcriptome were retained for initial differential expression comparisons but removed for clarity of presentation in subsequent analyses and volcano plots. Trimmed mean of M-values (TMM) (*Robinson and Oshlack, 2010*) was used for normalization and EdgeR (*Robinson et al., 2010*) was used to perform quasi-likelihood F-tests between A-P samples, corrected for multiple testing using FDR (Benjamini-Hochberg). Following filtering based on annotation and detection of >20 counts per paired samples, we used the following number of transcripts for differential expression comparisons: 5602 for *Clogmia*; 5392 for *Lutzomyia*; 8239 for *Culex*; 5802 for *Aedes*; 9353 for *Anopheles*; 5371 for *Nephrotoma*.

## Mapping RNA-seq reads to genomic loci

RNA-seq reads from stage-specific transcriptomes were mapped to genomic scaffolds containing a gene of interest using TopHat RNA-seq aligner (*Trapnell et al., 2009*). Publicly available *Anopheles stephensi* transcriptomes used in this paper were: SRR515316, SRR515341, SRR514863, and SRR515304.

## Data availability

This project was deposited at the National Center for Biotechnology Information under Bioproject ID PRJNA454000 and the reads were deposited in the Short Reads Archives under accessions SRR7132661, SRR7132662, SRR7132659, SRR7132660, SRR7132665, SRR7132666, SRR7132663 and SRR7132664 for *Clogmia*, SRR7134470, SRR7134469, SRR7134472, SRR7134471, SRR7134468, and SRR7134467 for *Lutzomiya*, SRR8729860, SRR8729859, SRR8729858, SRR8729857, SRR8729856 and SRR8729855 for *Anopheles*, SRR8729854, SRR8729853, SRR8729852 and SRR8729851 for *Aedes*, SRR8729868, SRR8729867, SRR8729870, SRR8729869, SRR8729864 and SRR8729863 for *Culex* and

SRR8729866, SRR8729865, SRR8729872, SRR8729871, SRR8729861 and SRR8729862 for *Nephrotoma*. Transcript sequences are listed on the Key Resources Table.

## Acknowledgements

We thank Chun Wai Kwan, Claudia Vacca, and Nicole Horio for technical assistance, Arthur Forer (York University, Canada) for the *Nephrotoma* culture, Robert Harrell and Channa Aluvihare (Insect Transformation Facility, University of Maryland, MD) for shipping blood-fed mosquitoes, Vanessa Macias and Anthony James (University of California at Irvine, CA) for *Aedes* and *Culex* reagents, Molly Duman Scheel (Indiana University, IN) for logistic support, and Edwin L Ferguson (University of Chicago, IL) and M Feder (University of Chicago, IL) for laboratory equipment. EL Ferguson provided detailed comments on a manuscript draft. This work was supported by funds from the National Science Foundation (IOS-1355057), the National Institute of General Medical Science (5R01GM127366-02), the University of Chicago, and the National Center for Advancing Translational Sciences of the National Institutes of Health (UL1 TR000430) to U S-O, and the Intramural Program of the National Institute of Allergy and Infectious Diseases to J R YY was the recipient of an award of University of Chicago Henry Hinds Funds for Graduate Student Research in Evolutionary Biology. Transcriptomic data are available at the National Center for Biotechnology Information Sequence Read Archive (Bioproject ID PRJNA454000). This work utilized the computational resources of the NIH HPC Biowulf cluster.

## Additional information

### Funding

| Funder | Grant reference number | Author |
| --- | --- | --- |
| National Science Foundation | IOS-1355057 | Urs Schmidt-Ott |
| National Institute of General Medical Sciences | R01 GM127366-01A1 | Urs Schmidt-Ott |
| National Center for Advancing Translational Sciences | UL1 TR000430 | Urs Schmidt-Ott |
| University of Chicago | Institutional fund | Urs Schmidt-Ott |
| National Institute of Allergy and Infectious Diseases | Intramural Program | Jose Ribeiro |
| University of Chicago | Henry Hinds Funds for Graduate Student Research in Evolutionary Biology | Yoseop Yoon |

The funders had no role in study design, data collection and interpretation, or the decision to submit the work for publication.

### Author contributions

Yoseop Yoon, Conceptualization, Formal analysis, Funding acquisition, Investigation, Visualization, Writing—original draft, Writing—review and editing; Jeff Klomp, Conceptualization, Formal analysis, Investigation, Visualization, Writing—original draft, Writing—review and editing; Ines Martin-Martin, Formal analysis, Investigation, Visualization; Frank Criscione, Investigation; Eric Calvo, Supervision; Jose Ribeiro, Resources, Formal analysis, Supervision, Funding acquisition; Urs Schmidt-Ott, Conceptualization, Supervision, Funding acquisition, Writing—original draft, Project administration, Writing—review and editing

### Author ORCIDs

Yoseop Yoon (iD) https://orcid.org/0000-0002-6062-0979
Urs Schmidt-Ott (iD) https://orcid.org/0000-0002-1351-9472

### Decision letter and Author response

Decision letter https://doi.org/10.7554/eLife.46711.038

Author response https://doi.org/10.7554/eLife.46711.039

## Additional files

### Supplementary files

• Supplementary file 1. Annotation and quantitation of *Clogmia* transcriptome. Gene names are those indicated as best-reciprocal-blast hits (see Methods). Contig names correspond to transcriptome assembly. Read counts are given based on non-unique mapping of pre-processed RNA-seq data to annotated transcriptome.
DOI: https://doi.org/10.7554/eLife.46711.026

• Supplementary file 2. Annotation and quantitation of *Lutzomyia* transcriptome. Gene names are those indicated as best-reciprocal-blast hits (see Methods). Contig names correspond to transcriptome assembly. Read counts are given based on non-unique mapping of pre-processed RNA-seq data to annotated transcriptome.
DOI: https://doi.org/10.7554/eLife.46711.027

• Supplementary file 3. Annotation and quantitation of *Culex* transcriptome. Gene names are those indicated as best-reciprocal-blast hits (see Methods). Contig names correspond to transcriptome assembly. Read counts are given based on non-unique mapping of pre-processed RNA-seq data to annotated transcriptome.
DOI: https://doi.org/10.7554/eLife.46711.028

• Supplementary file 4. Annotation and quantitation of *Aedes* transcriptome. Gene names are those indicated as best-reciprocal-blast hits (see Methods). Contig names correspond to transcriptome assembly. Read counts are given based on non-unique mapping of pre-processed RNA-seq data to annotated transcriptome.
DOI: https://doi.org/10.7554/eLife.46711.029

• Supplementary file 5. Annotation and quantitation of *Anopheles* transcriptome. Gene names are those indicated as best-reciprocal-blast hits (see Methods). Contig names correspond to transcriptome assembly. Read counts are given based on non-unique mapping of pre-processed RNA-seq data to annotated transcriptome.
DOI: https://doi.org/10.7554/eLife.46711.030

• Supplementary file 6. Annotation and quantitation of *Nephrotoma* transcriptome. Gene names are those indicated as best-reciprocal-blast hits (see Methods). Contig names correspond to transcriptome assembly. Read counts are given based on non-unique mapping of pre-processed RNA-seq data to annotated transcriptome.
DOI: https://doi.org/10.7554/eLife.46711.031

• Transparent reporting form
DOI: https://doi.org/10.7554/eLife.46711.032

### Data availability

Sequencing data have been deposited at the National Center for Biotechnology Information Sequence Read Archive (Bioproject ID PRJNA454000).

The following dataset was generated:

| Author(s) | Year | Dataset title | Dataset URL | Database and Identifier |
|---|---|---|---|---|
| Yoon Y, Klomp J, Martin-Martin I, Criscione F, Calvo E, Ribeiro J, Schmidt-Ott U | 2018 | Evolution of an Embryonic Axis Determinant via Alternative Transcription | https://www.ncbi.nlm.nih.gov/bioproject/PRJNA454000/ | NCBI Bioproject, PRJNA454000 |

The following previously published dataset was used:

| Author(s) | Year | Dataset title | Dataset URL | Database and Identifier |
|---|---|---|---|---|
| Xiaofang Jiang | 2012 | Anopheles stephensi strain:Indian | https://www.ncbi.nlm. | NCBI Bioproject, |

| Wild Type (Walter Reid) Transcriptome or Gene expression | nih.gov/bioproject?LinkName=sra_bioproject&from_uid=196910 | PRJNA168517 |

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
