## [Decision Letter]

Thank you for submitting your article "Evolution of embryonic axis determinants via alternative transcription" for consideration by *eLife*. Your article has been reviewed by three peer reviewers, and the evaluation has been overseen by Patricia Wittkopp as the Reviewing and Senior Editor. The following individuals involved in review of your submission have agreed to reveal their identity: Leslie Pick (Reviewer #1); Aleksander Popadic (Reviewer #2); Michael Akam (Reviewer #3).

The reviewers have discussed the reviews with one another and the Reviewing Editor has drafted this decision to help you prepare a revised submission.

Summary

Yoon et al. present a study of anterior determinants in Diptera, none of which utilize the *Drosophila* determinant, bcd. They identify opa, CG9215 and pangolin in different species through a combination of RNAseq, in situ hybridization, microinjection and RNAi in multiple species. Shared across genes and species, they find that alternate transcripts (alternate 3' or 5' ends) function in maternal and zygotic patterning. The first three figures present experiments on the moth fly, *Clogmia*. The authors convincingly demonstrate that opa functions as an anterior determinant in this species, analyzing expression and function in this species. They further show differential function of maternal and zygotic transcripts, which differ in expression and in 20 aa at the N-termini. Functions of the two isoforms are differentiated by RNAi, as well as ectopic injections, showing both loss and gain of function for this gene. The authors next demonstrate that opa coding sequences from other species are capable of functioning as anterior determinants in *Clogmia*. Next, the authors compare *Clogmia* opa to that of the sand fly *Lutzomyia*, finding alternate 5' and 3' UTRs in this species but no difference in protein coding regions of a maternal and zygotic transcript. Together, these results suggest that cis-regulatory evolution rather than protein evolution was the driver of opa taking on a role in anterior patterning. The authors next examine several mosquito species, in which opa is not maternally deposited (Figures 4 and 5). Rather, RNAseq for *Culex* identified CG9125 as an anteriorly localized gene that, when tested by RNAi, also appears to function as an anterior determinant. This gene also is differentially expressed and functions in *Aedes* mosquitoes. However, this gene was not detected as a potential anterior determinant in *Anopheles* mosquitoes and crane flies. Here, RNAseq identified pangolin as an anteriorly localized transcript. In all cases, alternate 5' ends were identified for maternal and zygotic transcripts. The paper also documents which of these species show posteriorly localised transcripts of the genes known to be involved in germ line specification. They identify such transcripts in all but the midge *Clogmia*, which appears not to localise any of the known germ cell determinants before blastoderm formation. This result is surprising, and another important finding of the paper. It perhaps deserves more prominence in the Abstract.

This is an impressive paper building on the established technical and conceptual expertise of the Schmidt-Ott group in working with axis specification in lower dipteran embryos. It extends work previously published in Science (Klomp et al., 2015), to generate remarkable further results. The study includes species of medical importance (*Aedes, Anopheles, Lutzomyia*) and work on a taxonomically significant group of basal dipterans, the crane flies, that have not previously been used for embryonic manipulation, to our knowledge. RNAseq analysis in bisected embryos was successful in identifying differentially localized gene products in both anterior and posterior regions of the embryo and led to the identification of novel anterior determinants in these species. The successful use of this approach is valuable for the field. Further, the authors present high quality in situ hybridization results in multiple species and, importantly, follow up on expression analysis with functional studies using both RNAi and injection of mRNA. These experiments appear to have been done rigorously and the figures are of very high quality. The data in Figure 2 particularly impressive.

Essential revisions

1) There was consensus among the reviewers that although the authors emphasize the theme of alternative transcripts, another key result from these experiments is the range of genes that can function as an anterior determinant in different species. The reviewers found this to be the more interesting and better supported conclusion of the work. As one reviewer wrote: The case for using these data as an evidence of AT is much less convincing, and it would need to be documented by rigorous testing. In addition, there is inconsistency in experimental design and over-reaching writing that detracts from the otherwise solid study. What the authors have been able to show, without any question, is that different proteins have been recruited to the anterior pole of the embryo to provide the anterior determination in different dipterans. But, this is not the same as documenting that this is due to alternative transcription. For example, in the best worked out species (*Clogmia*), the authors show that both maternal and zygotic opa transcripts can generate the same phenotype (double-headed larva), and consequently have the same function. In addition, the additional injections of mutated opa mRNA variants (Figure 3B) show that the obvious difference (additional 20aa in 5' end of zygotic transcript) has no effect on determining the anterior end. Surprisingly, it is the loss of 3'end sequence that has an effect. However, a closer inspection of the sequence alignment in Figure 1 supplemental shows that these regions are identical in both transcripts – hence the conundrum of explaining the difference in function. The only clue is provided in the Discussion "All anterior determinants that we report in this study contain either 5' or 3' UTR sequence that is not shared with the corresponding zygotic isoforms." But the authors do not provide any evidence in supplemental material to support this claim. If that information exists, can't that information be used to generate opa variants that can test the functional significance of these sequences? These types of experiments are required to generate evidence of alternative transcription. The paper suggests that in most cases the maternally expressed transcript is specific and likely newly evolved for this role. This is on the basis that (with one exception) the maternal transcripts are not expressed during the later stages of embryogenesis that have been examined. However, no data are shown to rule out the possibility that the maternal promoter is also expressed and functional at other life stages – for example, in the adult nervous system. It would be desirable to rule out this possibility, particularly if the authors want to stress the idea that these alternative transcripts really are novel inventions, and not simply the redeployment of an already established transcript isoform to a new role during oogenesis. The paper is written to strongly emphasise the fact that each of these novel anterior determinants is encoded by an alternative transcript form, and argues that this may be a common way for genes to acquire novel functions.

The claim about generality of AT as a mechanism and its significance is presented in the title, Abstract, Introduction, and Discussion, so there should be much stronger evidence in support of this claim. To show that AT is indeed responsible for the observed anterior axis determination the authors would have to commit to and perform rigorous experiments to show the differences in function between different transcripts, but these experiments would likely require more than two months to perform. We therefore encourage the authors to refocus the manuscript on the diversity of anterior determinants, removing prominent claims about alternative transcripts in the title and Abstract, at least. By their very nature as localised transcripts, these anterior determinants must contain specific localisation signals embedded in the RNA. This requirement may significantly increase the probability that such determinants evolve through alternative transcription, which allows novel DNA sequence to be expressed as RNA. It remains to be seen whether this would also be true for a randomly chosen set of novel gene functions. For that reason, and given the very considerable biological interest of the observation that novel determinants have evolved so frequently within the Diptera, we feel that stressing the message of alternative transcription might not be the best way to frame the paper.

2) The number of species used here, while quite impressive, is quite confusing for those outside the very immediate field. The species tree in Figure 1 has too little detail in Figure 1 while the one in Figure 6 has too much and doesn't indicate the species used in this paper. Perhaps a small table based on phylogenetic relationships among the species used would help? Photos of the species used? I would mention these relationships when each new species is introduced. For the purposes of this paper, those relationships are more important than the biomedical relevance of the species chosen.

3) Some experimental inconsistencies/over-interpretations were also noted. Some examples include: 1) probes used to detect expression patterns of the three Cqu-CG9215 transcripts (probe C is located in a conserved, shared region and is not specific to C-transcript only; it likely detects the expression of all three transcripts); 2) in Nephrotoma the authors have identified a single pangolin transcript that was labeled as Nsu-pan mat – the presence/absence of zygotic transcript has to be explicitly confirmed and stated as such; stating that on the basis of its expression "..localized maternal pangolin transcript functioned as anterior determinant in ancestral dipteran" is misleading; 3) similarly, stating that "For example, the anterior determinant of *Clogmia* (Cal-Opa^Mat^) lacks the N-terminal 20 amino acids (Figure 1—figure supplement 1), […] Such changes to the protein could have been important for adopting a function as anterior determinant." when in fact this difference was shown not to be significant (Figure 3B) is erroneous and misleading. Please correct any errors and re-check the full manuscript carefully to correct such cases.

Optional suggestion from one reviewer:

1) The discussion of germ plasm and gene expression at the posterior role is certainly of interest and evidence for inductive germ cell specification in flies would be of great interest. However, I find this information distracting, at least in its current form and placement within the paper. I have never before suggested to an author that they should remove data from a manuscript but I am thinking that these data might be better presented in a separate study.

[Editors' note: further revisions were requested prior to acceptance, as described below.]

Thank you for resubmitting your work entitled "A Range of Conserved Genes Establish Embryo Polarity in Moth Flies and Mosquitoes via Alternative Transcription" for further consideration at *eLife*. Your revised article has been favorably evaluated by Patricia Wittkopp as the Reviewing and Senior Editor, and three reviewers.

The manuscript has been improved but there are some remaining issues that need to be addressed before acceptance, as outlined below:

We all agreed that this study presents a very impressive body of work. The most strongly supported finding is that factors specifying the anterior end of the embryo are evolving more often than anticipated, and specific transcripts that play roles in this process are identified in moth flies and mosquito species. I think that this work will have a major impact on the field of evolution and development, and I look forward to seeing it published in *eLife*. However, there is one very important issue left to be resolved that requires only text changes.

This issue is the same as the primary issue raised during the first round of review: the framing of the work around alternative transcription (AT) is not justified by the strength of evidence presented that the AT was in fact responsible for the functional differences observed among species. All of the cases of AT described also include a difference in expression (presence and localization of the maternal transcript). The authors seem to be convinced that this difference in expression must be due to the differences in transcript structure (presumably differences in UTRs), but they do not do any experiments to directly demonstrate this (e.g., swapping UTRs between transcripts and showing that this causes the differences in localization, or expressing the zygotic transcript with the maternal promoter and showing this does not get localized similar to the maternal transcript). I realize that these are difficult experiments to perform in the species examined, so I am not suggesting that they be done. Rather, I’m saying that the conclusions need to be modified to reflect the fact these experiments haven't been done. For example, the different transcript structures might be a neutral byproduct of different promoters used to drive maternal and zygotic transcription. Without experiments disentangling AT from the expression differences, statements like " via Alternative Transcription" in the title, which implies causation, are inappropriate. Other examples of places where the strength of the evidence for AT are overstated include:

End of Abstract: "independently evolved the function of axis determinant via alternative transcription" (same issue as with title).

Discussion section: "All three genes are subject to alternative transcription." They are also all subject to differences in expression pattern, which should also be mentioned here.

In the response, the authors write "we wish to emphasize that AT of old genes provided opportunity for evolving the anterior determinant function." The problem is that AT facilitating this evolution isn't demonstrated. So, this is a fine possibility to raise in the discussion, but not appropriate for title and main conclusion in Abstract. Indeed, the response also says "Therefore, we propose that moth fly odd-paired evolved its specific function as axis determinant via a change in expression mediated by AT (maternal expression and localization), rather than in response to protein change.". Again, proposing this model in the Discussion is fine; presenting it the take home message of the paper in the title and Abstract is not.

In the response to reviewers, response 3 lists ways in which it was demonstrated that there are alternative transcripts with different expression patterns. But, the reviewers' question here wasn't whether evidence of AT was sufficiently strong, but rather whether the evidence that this AT was responsible for the new function (presumably by altering localization of the transcript) was sufficiently strong.

In other places in the manuscript, the statements made about AT are appropriate. For example:

End of introduction (with a slight tweak): "Our results show that a range of distinct old genes function as anterior determinant in different species by localizing alternative maternal transcript isoforms at the anterior egg pole. We therefore propose that AT “might have” played an important role in the evolution of this gene function and gene regulatory networks in fly embryos."

Results section: "We therefore propose that this gene function evolved via co-option when alternative maternal transcript of moth fly odd-paired became enriched at the anterior egg pole. " Wording here makes it clear this is a hypothesis/model.

Results section: "Taken together, our results suggest that cucoid acquired the anterior determinant function via the localization of a maternal transcript isoform with an alternative 3' end." This is agnostic to whether localization or AT causes the effect.

Results section: AT for pan: "Taken together with the isoform-specific transcript localization data presented above, these RNAi results support the hypothesis that pangolin acquired the anterior determinant function via the localization of a maternal transcript isoform with an alternative 3' end. " Good because it includes both the AT and expression difference.

Discussion section: "it is possible that AT facilitates the evolution of anterior determinants by providing the UTR sequence for isoform-specific localization signals that do not interfere with other gene functions" and "Additional experiments will be needed to test whether the unique UTR sequences of anterior determinants are essential for their localization at the anterior egg pole." A clear statement of the missing evidence to support the model. Good.

I think that changing the title more substantially will improve the paper. There are other important elements to this work that are not related to AT that will likely be missed by readers with a title and Abstract focused so specifically on AT. That is, I think the current framing of the work will reduce its impact on the field because it masks other important results such as the change in germ cell specification and the role of slp and mira in embryo polarity in *Clogmia*, which are not related to AT. I agree with the reviewers that these findings could make their own nice paper, but I defer to the author's preference to keep them in this work. Having dedicated section headings will help keep them from getting lost. Choosing a broader title less focused on AT (combined with mentioning these findings in the Abstract) would also help readers discover these results more readily. Perhaps something like "Divergent mechanisms of embryonic patterning (or polarity) among insects" would work for a title?

Finally, we think that reorganizing the Results section a bit to streamline it would improve readability of the manuscript (see reviewer comment below). However, I recognize that this is more subjective and leave it up to the author's discretion to decide whether and how to change the manuscript in response to this feedback.

Below are some of the comments from individual reviewers that elaborate on these concerns:

1) Insistence of interpreting data as a case of AT

In author's words "[...] Specifically, we provide evidence that the odd-paired ortholog of moth flies evolved without essential protein change an additional function as anterior determinant by maternally expressing and localizing transcript with alternative first exon in the anterior egg. This point is supported by RACE experiments and transcriptome profiling (Figure 1C), transcript-specific expression patterns (Figure 1D), transcript-specific functions (Figure 2), and comparative gain of function experiments (Figure 3)."

I tried to understand the authors' rationale, especially regarding the uppercase sentence that seems to suggest that it is an alternative first exon (present in Cal-otd mat transcript) that is responsible for the new function as anterior determinant. Data in Figure 1C shows the presence of two Cal-otd transcripts (Cal-Opa^Mat^ and Cal-Opa^Zyg^) in embryos and Figure 1D shows that these two transcripts have different expression patterns. This is followed by RNAi functional experiments that show that depletion of Cal-Opa^Mat^ results in double-abdomens, while Cal-Opa^Zyg^ knockdowns have defects in segmentation and dorsal closure (Figure 2A-B). So far, so good. Then we reach Figure 3D that beautifully illustrates that Cal-Opa^Mat^ and Cal-Opa^Zyg^ are interchangeable in their ability to function as an anterior determinant. In other words, this new function in A-P axis determination is not due to changes in the protein sequence! Instead, it is driven by temporal and spatial changes in regulation of otd expression. Most people would consider this an example of regulatory evolution, which is not associated with AT per se. On that note, it is unclear why authors imply that the first exon is responsible for the anterior determination function – this exon is absent in Cal-Opa^Zyg^, and yet this transcript can functionally substitute for Cal-Opa^Mat^.

2) Data presentation and logic flow

This study is rather complex, encompassing multiple species and genes and a number of different experimental approaches. The authors should keep this in mind and streamline their presentation and description of results. For example, the effects of Cal-Opa^Mat^ and Cal-Opa^Zyg^ RNAi are presented first in Figure 2A-B, to be followed by description Cal-nos expression in Figure 2A. Then, the text describes the effects of Cal-Opa^Mat^ mRNA injections in Figure 3A, to be followed by description of differential expression analysis in *Lutzomyia* (Figure 3B-C), then back to *Clogmia* and testing the effects of injections of mRNA of Cal-Opa^Mat^ and Cal-Opa^Zyg^ (Figure 3D), and so on.

A streamlined presentation of results would be greatly appreciated by general readership. This would entail keeping the current Figure 1 and text describing it, then modifying Figure 2 and 3 and reorganizing current text starting with Cal-Opa^Mat^ RNAi results (current Figure 2A bottom), followed by expression studies of selected marker genes (current Figure 2A top), and followed by results of Cal-Opa^Mat^ mRNA injections (current Figure 3A). This way, all functional experiments for Cal-Opa^Mat^ are seamlessly presented at one place. This can be followed by similarly presented results for Cal-Opa^Zyg^, and finished with results of mRNA injections of Cal-Opa^Mat^ and Cal-Opa^Zyg^ that highlight the functional interchangeability of two transcripts.

The structure and focus of the paper as re-submitted are similar to those of the first submission, despite the consensus view of the reviewers that it might usefully be changed. I think it unwise of the authors to ignore the reviewer's advice, but I do not think this a reason to reject the paper, so long as the specific concerns of the reviewers have been adequately addressed.

The Abstract is much improved and is now fine.

The structure of the Introduction remains not to my taste, but it is OK.

The new subheadings are useful, and new paragraphs providing clarification are generally helpful.

The handling of the germ cell data in a separate section is acceptable, though reviewer 1's suggestion to publish it elsewhere seemed to me to have merit.

---

## [Author Response]

Essential revisions1) There was consensus among the reviewers that although the authors emphasize the theme of alternative transcripts, another key result from these experiments is the range of genes that can function as an anterior determinant in different species. The reviewers found this to be the more interesting and better supported conclusion of the work. As one reviewer wrote: The case for using these data as an evidence of AT is much less convincing, and it would need to be documented by rigorous testing. In addition, there is inconsistency in experimental design and over-reaching writing that detracts from the otherwise solid study. What the authors have been able to show, without any question, is that different proteins have been recruited to the anterior pole of the embryo to provide the anterior determination in different dipterans. But, this is not the same as documenting that this is due to alternative transcription.

We agree that the range of genes that can function as an anterior determinant in different species is an interesting result of our study. However, our study also provides insight into the mechanism by which old genes evolve the anterior determinant function, and we would like to emphasize both. Specifically, we provide evidence that the *odd-paired* ortholog of moth flies evolved without essential protein changean additional function as anterior determinant by maternally expressing and localizing transcript with alternative first exon in the anterior egg. This point is supported by RACE experiments and transcriptome profiling (Figure 1C), transcript-specific expression patterns (Figure 1D), transcript-specific functions (Figure 2), and comparative gain of function experiments (Figure 3). Additionally, in Culex, only one of three alternative *cucoid (CG1925)* transcript isoforms with alternative last exon appears to be maternally expressed and localized (Figure 4B, C), and is therefore a plausible candidate of this gene’s anterior determinant function. Finally, in *Anopheles*, a *pangolin* transcript isoform with alternative last exon is maternally expressed and localized, and specifically required as anterior determinant (Figure 6B-D). The literature on AT is large but we are not aware of any comparative in vivostudy that better documents the new function of an alternative transcript isoform than this case. Therefore, we wish to emphasize that AT of old genes provided opportunity for evolving the anterior determinant function.

This is not the same as claiming that the anterior determinant function evolved “due to alternative transcription”, which on its own does not necessarily result in localized gene activity at the anterior egg pole. We realize that this difference was not made sufficiently clear in the original manuscript. To clarify this issue and to better weigh our results, we modified the title of the manuscript and carefully revised all sections of the manuscript, including the summaries provided in the Abstract, at the end of the Introduction, and at the beginning of the Discussion (see highlighted text in the comparison of the original and the revised manuscript texts).

Importantly, in the Results section, we added several subheadings and rationale to better define our goals in each species. In the revised version, results in *Culex* and *Aedes* are presented in separate figures (Figure 4 and Figure 5) because these two species serve slightly different purposes. Moreover, we moved the *Nephrotoma* results after the *Anopheles* data and clarify our specific goal with this species. Our goal with *Nephrotoma* has been to test whether anterior-localized maternal *pangolin* transcript is an old dipteran heritage. This was confirmed. We did not conduct a detailed analysis of alternative *pangolin* transcripts in this species because an assembled genome sequence is not available and because the purpose for including this species in the analysis was to test whether *pangolin* preceded *panish* as the anterior determinant gene.

For example, in the best worked out species (Clogmia), the authors show that both maternal and zygotic opa transcripts can generate the same phenotype (double-headed larva), and consequently have the same function. In addition, the additional injections of mutated opa mRNA variants (Figure 3B) show that the obvious difference (additional 20aa in 5' end of zygotic transcript) has no effect on determining the anterior end. Surprisingly, it is the loss of 3'end sequence that has an effect. However, a closer inspection of the sequence alignment in Figure 1 supplemental shows that these regions are identical in both transcripts – hence the conundrum of explaining the difference in function.

We revised the Results section to clarify our message. In the moth fly section, we provide evidence that the C-terminal sequence is important for Cal-Opa’s function as anterior determinant but this sequence could be important for any other function of Odd-paired as well. More surprisingly, we found that large N-terminal truncations do not preclude its role as anterior determinant (Figure 3D). In agreement with this observation, we propose that the natural occurrence of the N-terminal truncation in Cal-Opa^Mat^may not have played an important role in evolving the anterior determinant function. This hypothesis is supported by three additional lines:

· The zygotic ORF can function as anterior determinant if the corresponding transcript is presented in the pole region of very early embryos.

· The *odd-paired* ORFs of *Drosophila* and *Chironomus* can induce head development in *Clogmia*, if the corresponding transcript is presented in the pole region of very early embryos.

· The N-terminal truncation is not conserved in *Lutzomyia*, and the untruncated ORF of *Llo-opa^Mat^* can induce head development in *Clogmia*, if the corresponding transcript is presented in the pole region of very early embryos.

Therefore, we propose that moth fly *odd-paired* evolved its specific function as axis determinant via a *change in expression* mediated by AT (maternal expression and localization), rather than in response to protein change. This is a very important point because it provides insight into the *mechanism* by which moth fly *odd-paired* evolved its function as anterior determinant.

To put this in context, we would like to point out that a large body of work has been conducted to better understand how an axis determinant, such as *bicoid*, acquired its function (e.g., Datta et al., 2017; Liu et al., 2018). This work has been conducted under the premise that protein changes enabled Bicoid’s unique function as axis determinant, but this premise remains contentious (see last paragraph in the Discussion section of Liu et al., 2018). Our results in moth flies provide proof-of-concept for an alternative mechanism by which genes can evolve the anterior determinant function.

The only clue is provided in the Discussion "All anterior determinants that we report in this study contain either 5' or 3' UTR sequence that is not shared with the corresponding zygotic isoforms." But the authors do not provide any evidence in supplemental material to support this claim. If that information exists, can't that information be used to generate opa variants that can test the functional significance of these sequences? These types of experiments are required to generate evidence of alternative transcription.

Please also refer to response above. Based on our RACE, RNA-seq, and RNA in situ hybridization results, *Cal-opa* (Figure 1C, D), *cucoid* (Figure 4B, C) and *Aga-pan*golin (Figure 6B, C) are subject to alternative transcription. Information on non-overlapping UTR sequences is included in these figures and all sequences have been made publicly available (see accession numbers under Data availability).

In the case of *cucoid*, only probe C detected localized maternal transcript. This probe targets a region that is shared by all three *cucoid* transcripts and therefore, on its own, cannot establish isoform-specific transcription. However, probes specific to the two longer transcript variants (*cucoid^A^*and *cucoid^B^*; including the variant with complete ORF) only detected zygotic expression, indicating that those transcripts are not localized at the anterior pole of the egg. We therefore infer that of the three identified *cucoid* isoforms, only the very short very short *cucoid^C^* isoform is localized. Directly studying the expression of this isoform is challenging due to its short unique sequence (121 nucleotides). This rationale is explained in the main text in the paragraph summarizing our results in *Culex*.

The paper suggests that in most cases the maternally expressed transcript is specific and likely newly evolved for this role. This is on the basis that (with one exception) the maternal transcripts are not expressed during the later stages of embryogenesis that have been examined. However, no data are shown to rule out the possibility that the maternal promoter is also expressed and functional at other life stages – for example, in the adult nervous system. It would be desirable to rule out this possibility, particularly if the authors want to stress the idea that these alternative transcripts really are novel inventions, and not simply the redeployment of an already established transcript isoform to a new role during oogenesis.

We do not know if the localized isoforms evolved specifically for their role as anterior determinant. Additional RACE experiments in *Clogmia* failed to identify Cal-Opa^Mat^transcript in other tissues, including adult flies, provided that ovarian tissue was removed. However, it is conceivable that our experiments failed to detect transcript that was expressed at low level or in few cells.

In any case, it seems likely that the maternally expressed transcript isoforms are older than their ability to localize tightly at the anterior pole and to function as anterior determinant. In the revised manuscript, we emphasize that maternally expressed transcripts with alternative 5’ or 3’ end provided *opportunity* for evolving the anterior determinant function via transcript localization and co-option, irrespective of the evolutionary age and potential other uses of these transcript isoforms in other cell types.

The paper is written to emphasise strongly the fact that each of these novel anterior determinants is encoded by an alternative transcript form, and argues that this may be a common way for genes to acquire novel functions.The claim about generality of AT as a mechanism and its significance is presented in the title, Abstract, Introduction, and Discussion, so there should be much stronger evidence in support of this claim. To show that AT is indeed responsible for the observed anterior axis determination the authors would have to commit to and perform rigorous experiments to show the differences in function between different transcripts, but these experiments would likely require more than two months to perform. We therefore encourage the authors to refocus the manuscript on the diversity of anterior determinants, removing prominent claims about alternative transcripts in the title and Abstract, at least.

Please refer to our first response above.

By their very nature as localised transcripts, these anterior determinants must contain specific localisation signals embedded in the RNA. This requirement may significantly increase the probability that such determinants evolve through alternative transcription, which allows novel DNA sequence to be expressed as RNA.

We agree and have made this point at the end of the first paragraph of the Discussion section. The development of a suitable assay to map the transcript localization signal of *Cal-opa* is something we pursue but is not trivial in this species and beyond the goal of the present study.

It remains to be seen whether this would also be true for a randomly chosen set of novel gene functions. For that reason, and given the very considerable biological interest of the observation that novel determinants have evolved so frequently within the Diptera, we feel that stressing the message of alternative transcription might not be the best way to frame the paper.

Please refer to our first and second response above.

2) The number of species used here, while quite impressive, is quite confusing for those outside the very immediate field. The species tree in Figure 1 has too little detail in Figure 1 while the one in Figure 6 has too much and doesn't indicate the species used in this paper. Perhaps a small table based on phylogenetic relationships among the species used would help? Photos of the species used? I would mention these relationships when each new species is introduced. For the purposes of this paper, those relationships are more important than the biomedical relevance of the species chosen.

To address this issue, we made changes to the relevant figures (Figure 1A and Figure 7) and clarified the phylogenetic position of each species in the main text.

3) Some experimental inconsistencies/over-interpretations were also noted. Some examples include: 1) probes used to detect expression patterns of the three Cqu-CG9215 transcripts (probe C is located in a conserved, shared region and is not specific to C-transcript only; it likely detects the expression of all three transcripts);

This is correct. Probe C detects all three transcripts. However, probe C is the only probe that detected localized maternal expression of this gene and probes specific to the two longer transcript variants only detected zygotic expression, indicating that those transcripts are not localized at the anterior pole of the egg. Please refer to our third response above for additional details.

2) in Nephrotoma the authors have identified a single pangolin transcript that was labeled as Nsu-pan mat – the presence/absence of zygotic transcript has to be explicitly confirmed and stated as such; stating that on the basis of its expression "localized maternal pangolin transcript functioned as anterior determinant in ancestral dipteran" is misleading;

See also our first response above. We changed the labeling to *Nsu-pan* and revised the conclusion: “Taken together with our *Anopheles* data, our results in *Nephrotoma* suggest that ancestral dipteran insects localized maternal *pangolin* transcript in the anterior egg pole, where this transcript may have functioned as anterior determinant.”

*3) similarly, stating that "For example, the anterior determinant of Clogmia (*Cal-Opa^Mat^*) lacks the N-terminal 20 amino acids (Figure 1—figure supplement 1),.……. Such changes to the protein could have been important for adopting a function as anterior determinant." when in fact this difference was shown not to be significant (Figure 3B) is erroneous and misleading. Please correct any errors and re-check the full manuscript carefully to correct such cases.*

This quote is presented out of context. We present alternative hypotheses. To avoid misunderstanding, we slightly modified the relevant paragraph (second paragraph of the Discussion).

Optional suggestion from one reviewer:1) The discussion of germ plasm and gene expression at the posterior role is certainly of interest and evidence for inductive germ cell specification in flies would be of great interest. However, I find this information distracting, at least in its current form and placement within the paper. I have never before suggested to an author that they should remove data from a manuscript but I am thinking that these data might be better presented in a separate study.

We considered this possibility before submitting our manuscript. We decided against it because the question of a potential loss of the maternal germ plasm is raised by our profiling data. The loss of maternal germ plasm in *Clogmia* is supported by the duplication of *nos*-positive cells in double abdomens and adds confidence in the transcript localization data (Figure 1B). To address the reviewer concern, we now present these data under a separate subheading. This change gives readers the option to skip this section.

[Editors' note: further revisions were requested prior to acceptance, as described below.]

[…] This issue is the same as the primary issue raised during the first round of review: the framing of the work around alternative transcription (AT) is not justified by the strength of evidence presented that the AT was in fact responsible for the functional differences observed among species. All of the cases of AT described also include a difference in expression (presence and localization of the maternal transcript). The authors seem to be convinced that this difference in expression must be due to the differences in transcript structure (presumably differences in UTRs), but they do not do any experiments to directly demonstrate this (e.g., swapping UTRs between transcripts and showing that this causes the differences in localization, or expressing the zygotic transcript with the maternal promoter and showing this does not get localized similar to the maternal transcript). I realize that these are difficult experiments to perform in the species examined, so I am not suggesting that they be done. Rather, I’m saying that the conclusions need to be modified to reflect the fact these experiments haven't been done. For example, the different transcript structures might be a neutral byproduct of different promoters used to drive maternal and zygotic transcription. Without experiments disentangling AT from the expression differences, statements like " via Alternative Transcription" in the title, which implies causation, are inappropriate. Other examples of places where the strength of the evidence for AT are overstated include:End of Abstract: "independently evolved the function of axis determinant via alternative transcription" (same issue as with title).

We actually agree with the reviewers on this point. Maternal expression *and* anterior transcript localization (or activity localization) seem to be key for evolving the anterior determinant function. However, whether the localization signal or maternal expression came first (or together), and whether the localization signal is confined to transcript-specific sequence of the maternal isoform, may vary from case to case.

AT can accommodate the change in expression for evolving the anterior determinant function (maternal plus localized activity) by providing a promoter for maternal expression and/or by providing sequence for evolving a strong localization signal de novo, or for strengthening a more diffuse pre-existing localization signal. In each case, AT allows to minimize interference with pre-existing gene regulation and function when a new anterior determinant evolves. But this is not to say that the localization signals of anterior determinants must be confined to transcript-specific sequence of the maternal isoform.

In this context, we would like to point out that the transcripts of many pair-rule genes of *Drosophila* contain localization signals of variable length, and would likely be localized like *bicoid* if expressed maternally, because they use the same microtubule-dependent machinery (e.g., Bullock, Stauber … and Schmidt-Ott, 2004, and other work from Bullock et al.). Conversely, *bicoid* mRNA is localized apically like pair-rule gene transcripts when injected into the syncytial blastoderm.

To avoid being misunderstood, we removed the expression “via alternative transcription” in the title, which now reads: “Embryo polarity in moth flies and mosquitoes relies on distinct old genes with localized transcript isoforms”. Additionally, we revised the Abstract and conclude: “In conclusion, flies evolved an unexpected diversity of anterior determinants, and alternative transcript isoforms with distinct expression can adopt fundamentally distinct developmental roles.” The latter point is mainly based on our *Clogmia* results but seems important to us, because the neglect of alternative transcription in the evo devo literature suggests that this possibility is not widely appreciated in this field.

Discussion section: "All three genes are subject to alternative transcription." They are also all subject to differences in expression pattern, which should also be mentioned here.

The sentence appears in the first paragraph of the Introduction and was changed to: “All three genes not only localize their maternal transcript at the anterior egg pole; they also are subject to alternative transcription, which allows a single gene to generate multiple transcript isoforms with distinct 5’ and 3’ ends through the use of alternative promoters (alternative transcription initiation) and polyadenylation signals (alternative transcription termination).”

The additional explanation was necessary, because we shifted the section on alternative transcription from the Introduction to the Discussion.

In the response, the authors write "we wish to emphasize that AT of old genes provided opportunity for evolving the anterior determinant function." The problem is that AT facilitating this evolution isn't demonstrated. So, this is a fine possibility to raise in the discussion, but not appropriate for title and main conclusion in Abstract. Indeed, the response also says "Therefore, we propose that moth fly odd-paired evolved its specific function as axis determinant via a change in expression mediated by AT (maternal expression and localization), rather than in response to protein change." Again, proposing this model in the discussion is fine; presenting it the take home message of the paper in the title and Abstract is not.

We moved the introductory paragraphs on alternative transcription, in a condensed form, to the Discussion section.

In the response to reviewers, response above lists ways in which it was demonstrated that there are alternative transcripts with different expression patterns. But, the reviewers' question here wasn't whether evidence of AT was sufficiently strong, but rather whether the evidence that this AT was responsible for the new function (presumably by altering localization of the transcript) was sufficiently strong.

We do not wish to claim that AT per se was responsible for the new function, but AT could have facilitated the evolution of localization signals and/or maternal expression without interfering with pre-existing gene regulation/function (as discussed under response above).

In other places in the manuscript, the statements made about AT are appropriate. For example:End of introduction (with a slight tweak): "Our results show that a range of distinct old genes function as anterior determinant in different species by localizing alternative maternal transcript isoforms at the anterior egg pole. We therefore propose that AT “might have” played an important role in the evolution of this gene function and gene regulatory networks in fly embryos."

The end of the Introduction now concludes with: “Our results reveal three distinct old genes that evolved anterior determinants by localizing an alternative maternal transcript isoform at the anterior egg pole of the respective species. Therefore, alternative transcription might have played an important role in the evolution of this gene function and gene regulatory networks in fly embryos.”

Results section: "We therefore propose that this gene function evolved via co-option when alternative maternal transcript of moth fly odd-paired became enriched at the anterior egg pole. " Wording here makes it clear this is a hypothesis / model.Results section: "Taken together, our results suggest that cucoid acquired the anterior determinant function via the localization of a maternal transcript isoform with an alternative 3' end." This is agnostic to whether localization or AT causes the effect.Results section: AT for pan: "Taken together with the isoform-specific transcript localization data presented above, these RNAi results support the hypothesis that pangolin acquired the anterior determinant function via the localization of a maternal transcript isoform with an alternative 3' end. " Good because it includes both the AT and expression difference.Discussion section: "it is possible that AT facilitates the evolution of anterior determinants by providing the UTR sequence for isoform-specific localization signals that do not interfere with other gene functions" and "Additional experiments will be needed to test whether the unique UTR sequences of anterior determinants are essential for their localization at the anterior egg pole." A clear statement of the missing evidence to support the model. Good.

Thank you for this helpful feedback.

I think that changing the title more substantially will improve the paper. There are other important elements to this work that are not related to AT that will likely be missed by readers with a title and Abstract focused so specifically on AT. That is, I think the current framing of the work will reduce its impact on the field because it masks other important results such as the change in germ cell specification and the role of slp and mira in embryo polarity in Clogmia, which are not related to AT. I agree with the reviewers that these findings could make their own nice paper, but I defer to the author's preference to keep them in this work. Having dedicated section headings will help keep them from getting lost.

We added the following sentence in the Abstract: “Additionally, *Clogmia* lost maternal germ plasm, which contributes to embryo polarity in fruit flies (*Drosophila*).” In the Results section, we discuss these data at the end of the *Clogmia* section under the sub-heading: “Cal-Opa^Mat^ suppresses zygotic germ cell specification at the anterior pole and *Clogmia* lacks maternal germ plasm”.

Choosing a broader title less focused on AT (combined with mentioning these findings in the Abstract) would also help readers discover these results more readily. Perhaps something like "Divergent mechanisms of embryonic patterning (or polarity) among insects" would work for a title?

As mentioned above, we revised the title: “Embryo polarity in moth flies and mosquitoes relies on distinct old genes with localized transcript isoforms”. We hope this title is acceptable. It focuses on facts that we wish to stress but does not infer causality. The title that you suggest would apply equally well to several previous studies on axis specification in *Tribolium, Nasonia*, and *Chironomus*, and would fail to specify what is new about the present study. Our study is the first to show that the anterior determinant function can evolve via change in expression, only.

Finally, we think that reorganizing the Results section a bit to streamline it would improve readability of the manuscript (see reviewer comment below). However, I recognize that this is more subjective and leave it up to the author's discretion to decide whether and how to change the manuscript in response to this feedback.

We essentially followed the reviewer suggestions (2). Specifically, we removed the paragraphs on AT in the Introduction to the last section of the Discussion. We wish to discuss our results in the broader context of this literature because, to our knowledge, there is a lack of well documented examples in which alternative transcripts of a gene have taken on clearly distinct developmental roles.

Importantly, we also reorganized the Result section, keeping the *Clogmia* data together, as suggested by the reviewer. Accordingly, some main figure compositions, and the numbering of some supplementary figures, have changed in the revised manuscript. In the revised manuscript, the *Clogmia* section ends with the loss of maternal germ plasm in this species. We think that it is important to include these data, not only because the volcano plot raises this issue, but also because germ plasm has been implicated in axis specification (e.g., Nos-dependent maternal Hb gradient in *Drosophila* embryos). To clarify the rationale for including this section we made slight changes to this section and added two references that established the role of the maternal Hb gradient in axis specification in *Drosophila* (Tautz, 1988 and Struhl et al., 1992).

We did not expand the section of *sloppy-paired* and *miranda* transcripts in *Clogmia*, because the localization of these transcripts does not seem to be essential for axis specification (see gain of function experiments with *Cal-opa*), and because their localization is not conserved in the other moth fly, *Lutzomyia*, unlike the localization of *odd-paired* transcript.

Below are some of the comments from individual reviewers that elaborate on these concerns:1) Insistence of interpreting data as a case of ATIn author's words "[…] Specifically, we provide evidence that the odd-paired ortholog of moth flies evolved without essential protein change an additional function as anterior determinant by maternally expressing and localizing transcript with alternative first exon in the anterior egg. This point is supported by RACE experiments and transcriptome profiling (Figure 1C), transcript-specific expression patterns (Figure 1D), transcript-specific functions (Figure 2), and comparative gain of function experiments (Figure 3)."

*I tried to understand the authors' rationale, especially regarding the uppercase sentence that seems to suggest that it is an alternative first exon (present in Cal-otd mat transcript) that is responsible for the new function as anterior determinant. Data in Figure 1C shows the presence of two Cal-otd transcripts (*Cal-Opa^Mat^*and* Cal-Opa^Zyg^*) in embryos and Figure 1D shows that these two transcripts have different expression patterns. This is followed by RNAi functional experiments that show that depletion of* Cal-Opa^Mat^*results in double-abdomens, while* Cal-Opa^Zyg^
*knockdowns have defects in segmentation and dorsal closure (Figure 2A-B). So far, so good. Then we reach Figure 3D that beautifully illustrates that* Cal-Opa^Mat^*and* Cal-Opa^Zyg^
*are interchangeable in their ability to function as an anterior determinant. In other words, this new function in A-P axis determination is not due to changes in the protein sequence! Instead, it is driven by temporal and spatial changes in regulation of otd expression. Most people would consider this an example of regulatory evolution, which is not associated with AT per se.*

We do indeed propose that regulatory evolution was key. Maternal expression and transcript localization (gene activity localization) had to be achieved for *odd-paired* to adopt the role of anterior determinant. In the case *Cal-opa*, the maternal promoter resulted from AT, and we suspect that the Cal-Opa^Mat^ -specific 5’UTR (also a result of AT) provided opportunity for evolving a transcript-specific localization signal after maternal expression. Alternatively, the localization signal was already in place (anywhere in the *Cal-opa* transcript) and evolved before maternal expression. A combination of both models is also possible. The links of AT to transcript localization can be manifold, as outlined under response one above, and probably provide yet another example of opportunistic evolution. What we wish to make clear is that AT is a common theme in the evolution of anterior determinants but its specific use varies case by case.

*On that note, it is unclear why authors imply that the first exon is responsible for the anterior determination function – this exon is absent in* Cal-Opa^Zyg^*, and yet this transcript can functionally substitute for* Cal-Opa^Mat^.

In the gain-of-function experiments, we tested mRNAs with heterologous UTRs provided by the vector, i.e., only the open reading frames of these transcripts were compared under the same artificial “localization” conditions (mRNA injection at the posterior pole). The distinct open reading frames of Cal-Opa^Mat^ and Cal-Opa^Zyg^ were functionally indistinguishable in our assay, indicating that the alternative transcription start site and/or unique 5’UTR of Cal-Opa^Mat^ (both a result of AT) were key for evolving the anterior determinant function of *odd-paired* in moth flies.

2) Data presentation and logic flow

*This study is rather complex, encompassing multiple species and genes and a number of different experimental approaches. The authors should keep this in mind and streamline their presentation and description of results. For example, the effects of* Cal-Opa^Mat^*and* Cal-Opa^Zyg^
*RNAi are presented first in Figure 2A-B, to be followed by description Cal-nos expression in Figure 2A. Then, the text describes the effects of* Cal-Opa^Mat^*mRNA injections in Figure 3A, to be followed by description of differential expression analysis in Lutzomyia (Figure 3B-C), then back to Clogmia and testing the effects of injections of mRNA of* Cal-Opa^Mat^*and* Cal-Opa^Zyg^
*(Figure 3D), and so on.*

*A streamlined presentation of results would be greatly appreciated by general readership. This would entail keeping the current Figure 1 and text describing it, then modifying Figure 2 and 3 and reorganizing current text starting with* Cal-Opa^Mat^*RNAi results (current Figure 2A bottom), followed by expression studies of selected marker genes (current Figure 2A top), and followed by results of* Cal-Opa^Mat^*mRNA injections (current Figure 3A). This way, all functional experiments for* Cal-Opa^Mat^*are seamlessly presented at one place. This can be followed by similarly presented results for* Cal-Opa^Zyg^*, and finished with results of mRNA injections of* Cal-Opa^Mat^*and* Cal-Opa^Zyg^
*that highlight the functional interchangeability of two transcripts.*

The structure and focus of the paper as re-submitted are similar to those of the first submission, despite the consensus view of the reviewers that it might usefully be changed. I think it unwise of the authors to ignore the reviewer's advice, but I do not think this a reason to reject the paper, so long as the specific concerns of the reviewers have been adequately addressed.The Abstract is much improved and is now fine.The structure of the Introduction remains not to my taste, but it is OK.The new subheadings are useful, and new paragraphs providing clarification are generally helpful.The handling of the germ cell data in a separate section is acceptable, though reviewer 1's suggestion to publish it elsewhere seemed to me to have merit.

We are grateful for these comments and implemented them as described under response one above.